# Comprehensive Survey of Consensus Docking for High-Throughput Virtual Screening

**DOI:** 10.3390/molecules28010175

**Published:** 2022-12-25

**Authors:** Clara Blanes-Mira, Pilar Fernández-Aguado, Jorge de Andrés-López, Asia Fernández-Carvajal, Antonio Ferrer-Montiel, Gregorio Fernández-Ballester

**Affiliations:** Instituto de Investigación, Desarrollo e Innovación en Biotecnología Sanitaria de Elche (IDiBE), Universidad Miguel Hernández, Av. de la Universidad s/n, 03202 Elche, Spain

**Keywords:** molecular docking, virtual screening, consensus docking, binding site, scoring function, drug discovery

## Abstract

The rapid advances of 3D techniques for the structural determination of proteins and the development of numerous computational methods and strategies have led to identifying highly active compounds in computer drug design. Molecular docking is a method widely used in high-throughput virtual screening campaigns to filter potential ligands targeted to proteins. A great variety of docking programs are currently available, which differ in the algorithms and approaches used to predict the binding mode and the affinity of the ligand. All programs heavily rely on scoring functions to accurately predict ligand binding affinity, and despite differences in performance, none of these docking programs is preferable to the others. To overcome this problem, consensus scoring methods improve the outcome of virtual screening by averaging the rank or score of individual molecules obtained from different docking programs. The successful application of consensus docking in high-throughput virtual screening highlights the need to optimize the predictive power of molecular docking methods.

## 1. Introduction

The process of discovering new drugs for the treatment of diseases includes the selection of appropriate targets, the identification of hits, and their optimization to increase the affinity, specificity, efficacy, metabolic stability, and oral bioavailability. When a compound that fulfills all these requirements is identified, drug development continues, and clinical trials are carried out to validate their therapeutic value. Despite advancements in resources and technologies, drug discovery still remains as a long, arduous, and expensive process. Dhasmana et al. refer to an average of 10 years and a considerable investment to develop a new drug [1]. This is mainly due to the high attrition rate in the clinical success of therapeutic agents [2]. To improve the success rates of drug discovery, new technologies with higher precision are demanded. In fact, the reduction of attrition in early stages of drug discovery is of capital importance to avoid costly failures of poor performance in late stages [3] due to the inaccurate selection of drug targets or inaccurate identification of leads, or both.

Experimental high-throughput screening (HTS) is commonly used to screen huge libraries of compounds to discover hits targeting the desired biological activity, which is the basis of the drug development. For example, automated patch-clamp or microfluorography are HTS techniques commonly used to speed up the screening of compound libraries on ion channels. The automated patch-clamp has recently evolved to improve the efficiency of the seals on the cell surface as well as the perfusion system, although its performance is still low. Microfluorography uses fluorescent dyes to monitor variations of ion concentrations in the cytosol or the changes in membrane voltage as a consequence of ion channel activity but has a low resolution and a high false positive rate. Quantitative HTS complements both approaches, obtaining dose–response curves by testing compounds at different concentrations. However, HTS does not generate lead compounds as quickly as desired [4] due to known limitations in the screening process, such as problems with aggregation, solubility, target expression systems, etc. [5]. The failure of HTS methods has greatly boosted the development of rational-based approaches to screen millions of molecules, which avoid the limitations of experimental HTS.

It has been previously recognized that the use of computer-assisted drug discovery methods greatly reduces costs if the binding affinities can be accurately predicted prior to performing the experiments. Thus, the employment of computational methods strongly increases the efficiency in the development of new compounds [6]. In this sense, the computational strategy has revolutionized rational drug design. The advantages of in silico approaches include high speed, low cost, automatization, and nearly unlimited scalability. Nonetheless, computational approaches require high-resolution 3D structures and methods to measure the theoretical binding energy of dozens of potential molecule conformations within the binding pocket. By far, the measurements of the binding affinity of thousands of compounds on a given target are the most laborious and costly task in drug discovery.

Computer-assisted methods are traditionally categorized in ligand-based and structure-based drug discovery [7] (Figure 1). Ligand-based methods require active ligands and use quantitative structure–activity relationship (QSAR) models, pharmacophore, and chemical similarity to predict new compounds [8,9]. Structure-based methods require the 3D structure of the receptor and ligands, and the discovery of new active compounds is based on the determination of physical interactions between the receptor and small molecules to form a biologically active complex [10]. The comprehension of the binding mechanism between receptor and ligands is crucial for drug discovery and optimization. In fact, the identification of the binding sites and the description of ligand–receptor interactions at the atomic scale are the main goals of structure-based methods, and much effort has been made to improve all these protocols [11] with molecular docking and virtual screening (VS) procedures being the most used procedures by far. Due to the rapid development of crystallography, NMR, cryo-electron microscopy (cryo-EM), and homology modeling, the structure-based VS technique has emerged as a useful technique for identifying potential hits during the early stage of drug discovery.

In this review, we will briefly revise computational approaches commonly used for the high-throughput VS approach, with special attention paid to molecular docking and consensus scoring. The performance of isolated docking programs is discussed, as well as the advantage of using multiple docking programs and different consensus strategies to globally optimize docking results, which has been proven to be particularly efficient in VS campaigns. A case study of the vanilloid receptor (TRPV1) and a library of known inhibitors and decoys docked to the vanilloid binding site is also included and discussed.

## 2. Structural Data Determination

Knowledge of the atomic structure of a protein or a closely related protein is pivotal when using computer-based strategies for the rational design of ligands acting as effectors. Computational techniques are more accurate if the protein target is known at the atomic level, including the binding pocket, where it is preferably occupied by an inhibitor or activator. The traditional technique to solve protein structures is X-ray crystallography, and there are currently nearly 150,000 entries shared in the Protein Data Bank (PDB), which represents around 76% of the total number of structures deposited in the database (as of October 2022). In this technique, diffracting crystals are needed to build accurate structural models where the 3D spatial position of atoms is determined. The X-ray technique displays major limitations for proteins that have difficulties in expression, purification, and/or crystallization, with membrane proteins being the most affected. This is a serious drawback for drug discovery as membrane proteins, and more precisely ion channels, are usually recognized as main therapeutic targets for human disorders such as metabolic, gastrointestinal, respiratory, cardiovascular, immunological, cancer, pain, and infectious diseases [12,13].

To overcome these limitations, the cryo-EM technique has represented a true revolution to the field of membrane protein structures. The technique takes images of grids containing, e.g., the ion channel frozen, and obtains the projection of the protein molecules by slowly rotating the grids in all spatial directions during imaging. The 2D images are sorted, aligned, and computed to reconstruct the 3D structure of the protein [14]. In addition, the development and use of nanodiscs has allowed rendering the membrane protein stable, inserted in a native bilayer with controlled composition [15], which is amenable to be studied by cryo-EM. The determination of protein structures with the cryo-EM approach benefits from the following facts: protein crystals are no longer needed [16]; it can be used for large protein complexes, including effectors; it maintains the functional states of the protein; and it is able to determine multiple conformational states of proteins in a single experiment [13]. On the contrary, limitations of the technique imply low spatial resolution structures (currently around 3 Å), and potential protein damage due to the low temperature and high radiation used during imaging. Currently, there are more than 12,500 structures determined by the cryo-EM technique in the PDB (by October 2022), and it represents a promising tool to cover the needs of targets for drug discovery.

Despite all of these advancements, the current entries in the PDB represent a very small fraction of the huge amounts of known proteins whose structures have not yet been resolved. To obtain access to these structures, several computational prediction methods are available. Homology modeling is the most used method to construct a reliable model for a protein whose structure is not deposited in the PDB; it is based on the selection of a homologous structure (the template) with a high identity and similarity with the chosen protein (the target), assuming that the protein structure is more conserved than the sequence. Then, the backbone conformation is transferred from the template and the side chains are assigned according to the sequence target, where they are selected from a library of rotamers. Accurate homology models are usually built up in web portals or in standalone programs, which are then refined to produce optimal homology models [17]. In this process, eventual effectors (ligands, inhibitors, cofactors, prosthetic groups, etc.) are kept in their pockets for further use [18], improving the geometry of the model. There are many web servers and programs for homology modeling, such as MODELLER, SWISS-MODEL, Rosetta, HHpred, and I-TASSER (see Fernandez-Ballester et al. [13]). In addition, because integral membrane proteins are embedded in lipid bilayers, sometimes it is necessary to model lipids around the protein and water to simulate an external and cytosolic environment [19,20]. Examples of programs used to prepare the initial configuration of these receptors are the CHARMM-GUI membrane builder web application (https://www.charmm-gui.org/; accessed on 7 November 2022) or the standalone program VMD [21].

Recent advances in ab initio computational structure prediction have catapulted AlphaFold as the most accurate tool for protein folding prediction [22,23]. AlphaFold is an artificial intelligence (AI) system developed by DeepMind that directly predicts the 3D structure of a protein from its amino acid sequence and aligned sequences of homologs. It incorporates a novel neural network and training procedures based on geometrical and physical constraints to improve the accuracy of the structure prediction [22]. The AlphaFold Protein Structure Database (https://alphafold.ebi.ac.uk/; accessed on 7 October 2022) provides access to nearly 200 million protein structure predictions and 48 complete proteomes (by October 2022), including sequences from the “one sequence per gene” reference proteome provided in UniProt 2021_04 (https://www.uniprot.org/release-notes/2021-11-17-release; accessed on 11 October 2022). Furthermore, Meta AI has predicted the structure of 600 million proteins from bacteria, viruses, and other microorganisms that have not been previously characterized. Meta’s network is not as accurate as AlphaFold, but it is over 60 times faster (https://www.nature.com/articles/d41586-022-03539-1; accessed on 15 November 2022). Similarly, RoseTTAFold is a “three-track” neural network to predict protein structures based on limited information. It considers patterns in protein sequences, amino acids interactions between proteins, and a possible three-dimensional structure to allow the network to decide the relationship between sequence and folding [24].

It has been shown that either homology or ab initio models allow for effective virtual screening. Several studies have been carried out comparing the performance of homology models and X-ray crystal structures of, e.g., G-protein coupled receptors. Carlsson et al. compared the virtual screening results obtained using these scaffolds and showed that the homology model was as effective as the crystal structure at detecting active ligands in terms of hit rate detection, potency, and novelty [25]. Similarly, Lim et al. found that 10 out of 19 G-protein coupled receptor homology models presented better or comparable performance than the corresponding crystallographic structures, making homology models suitable for virtual screening. They also explored consensus enrichment across multiple homology models, obtaining results comparable to the best performing model, highlighting the usefulness of the consensus scores [26]. Regarding AlphaFold, several studies have confirmed the suitability of these models to perform reliable VS campaigns. Wong et al. used 12 essential proteins, 218 active compounds, and 100 inactive compounds to predict antibacterial inhibitors and found that, although models had low performance, the use of rescoring strategies may have acceptable predictive power for certain proteins. They concluded that the limitations in benchmarking are not due to the AlphaFold structures itself, but to the methods to accurately model the protein–ligand interactions [27]. Other studies have identified potential inhibitors of WD40 repeat and SOCS box containing 1 protein (WSB1), a clinically relevant drug target, by means of AlphaFold and virtual screening approaches [28].

## 3. Computational Approaches Based on Structural Data: Protein Docking

Molecular docking predicts the interactions between a small molecule (ligand) and a protein binding site (receptor). The approach helps to identify the binding conformation and orientation of ligands in the binding pocket of receptors, thus determining the mechanisms of drug binding to targets. The process mimics the lock-and-key model of drug action to infer shape complementarity and affinity of a ligand within the binding site. Ligands can be organic molecules, peptides, and proteins [29,30]. Docking has evolved to study the principles of ligand–receptor molecular recognition. Nevertheless, due to the potential of docking in drug discovery programs, a great effort has been made to enhance the performance and accuracy of algorithms [31]. A successful docking accounts for ligand flexibility at different degrees (Figure 2), often distinguishing several levels of simplification [30].

Rigid docking refers to the fact that the conformation of the ligand and receptor do not change during the coupling process (Figure 2A), which is used in large systems such as protein–protein or peptide–protein docking. The algorithm evaluates different poses, and the best score is selected [32]. On the contrary, in flexible docking, both the receptor and ligand are free to change their conformations, which demands huge computational resources (Figure 2D). In the case of small peptides or organic molecules, the most common docking protocols involve ligands to be free to move, and the receptor is either rigid (Figure 2B) or only moves a few side chains in the binding pocket (Figure 2C) to improve the ligand–receptor coupling. This last interaction is called semi-flexible docking, which allows limited conformational changes either in the receptor and/or the ligand. A nice example is MedusaDock [33], which contemplates the possibility of changing the ligand and receptor at the same time with sets of discrete rotamers.

Molecular docking software is based on two basic pillars: a conformational search algorithm and a scoring function.

### 3.1. Search Algorithms

Search algorithms explore the optimal conformation of ligands within the pocket. The methods for sampling ligand conformations are usually defined as (i) systematic, (ii) simulation search, and (iii) stochastic search methods. The systematic algorithms explore all degrees of freedom in the molecule, including exhaustive search, fragment growth methods [34], or multiple conformer generation [35], which are individually docked against the target. The simulation search uses the solutions to Newton’s equations of motion for molecular dynamics or energy minimization [30]. The most widely used stochastic search algorithms perform random changes in the molecule to explore the conformational space, the most popular being the Monte Carlo, tabu search, swarm optimization, and genetic algorithm [36]. The Monte Carlo algorithm randomly generates small changes in the orientation, position, or ligand conformation to generate poses that must be accepted or rejected according to the Metropolis criterion [37]. At the beginning, the conformational freedom is high so that the probability of acceptance of ascending steps of energy is high, which facilitates the escape of energy traps. Then, the conformational freedom progressively decreases to find a low energy state of the ligand–receptor complex. The tabu search employs local search methods for conformational optimization; this method accepts worsening movements at the beginning if no improving moves are available. Additionally, it introduces prohibitions to avoid previously visited solutions, so that the conformations are no longer visited. The genetic algorithm is inspired by the Darwin’s theory of evolution. In this algorithm, the starting ligand conformation and orientation (parents) are changed to produce the second generation of conformations (descendants). The best-ranked energy conformations are used to produce the next generation [38].

In the case of highly flexible ligands such as peptides, there are considerable differences between the bound and unbound ligand structures [29]. In this case, an adaptation of the sampling strategy is needed to screen a large conformational space. It is common to use constraints derived from experimental data (biological, NOE, etc.) to limit the number of degrees of freedom [39].

### 3.2. Scoring Functions

The scoring function that evaluates protein–ligand binding is the most critical component of the docking method, and therefore must be robust, accurate, and fast. [40]. The scoring functions are usually classified in several categories [41]: (i) force field methods, (ii) empirical scoring functions, (iii) knowledge-based potentials, and (iv) machine learning scoring functions. Force field-based scoring functions use force fields, a collection of fundamental molecular terms that evaluate van der Waals, coulombic, and desolvation interactions between and within interacting molecules [42]. The equations and associated constants are derived from experimental data or quantum mechanics calculations. Entropy and desolvation terms are usually ignored or oversimplified. Poisson–Boltzmann or generalized Born equations are used to compute the desolvation energy of ligands [43]. Empirical scoring functions calculate the binding affinity of ligands and receptors based on weighted terms similar to force field scoring functions, including interaction types such as hydrophobic, hydrogen bonds, electrostatic, van der Waals, desolvation, entropy, etc. The coefficients of each term (the weight) are fitted using multiple linear regression from the training data [44]. The knowledge-based scoring functions, also known as the potential of mean force scoring functions, are statistical potentials derived from the study of protein–ligand structures deposited in the databases, and they are used as a training set. The method computes the frequency of occurrence of interacting atom pairs in receptors and ligands and generates the potentials using an inverse Boltzmann distribution [45]. Machine learning scoring functions use descriptors of known ligand–receptor interactions to build a machine learning model to derive a non-linear energy functional form of the binding affinity. Several machine learning algorithms are commonly used, such as the support vector machine [46], deep convolutional neural network, graph neural network [47] or random forest [48] algorithms to derive the machine learning scoring functions. These functions for the prediction of binding affinity have experienced large improvements in recent years, as recently reviewed by Yang et al. [40].

Most docking software uses generic scoring functions which usually report extensive validation test upon publication, demonstrating their superior performance. It should be pointed out that these functions handle targets unevenly due to certain chemical and structural features, including the size or exposure of the binding site, the presence of charged groups, or the presence of the cofactors/ion metals near the binding site, as well as the protonation state, partial charges, and number of rotatable bonds. Thus, it is almost impossible to anticipate the best scoring function for a given target, and the choice commonly relies upon the availability of the docking software implementing this or another function. The selection of a specific scoring function for a given target involves the design and optimization of a dataset of actives/decoys for the specific target. This strategy is subject to the availability of experimental information but allows for a clear and quantitative definition of the limit of validity of the different scoring functions by testing the selected library compounds on the binding site of the selected target [49].

Although machine learning scoring functions have shown superior performance compared with classical methods, a cloud of doubts hangs on these scoring functions due to its poor generalization capability and the unfair evaluation of the environment [50]. In this sense, the imbalanced datasets, dataset partitioning, or hidden data biases need to be handled for specific targets. Wallach et al. proposed the asymmetric validation embedding (AVE) strategy to decrease the effects of hidden biases and to avoid similarities between validation and training datasets, which represents an important tool to evaluate the general applicability of the scoring function based on machine learning [51].

## 4. Computational Approaches Based on Structural Data: Virtual Screening (VS)

High-throughput virtual screening (HTVS) identifies long lists of chemical structures predicted for putative binding to a protein target with high affinity. The structure-based VS method employs search algorithms to find optimal interactions between ligands and receptors and evaluates the affinity of the ligand–receptor complex [52]. In this way, the VS method obtains a long list of molecules ranked according to their binding scores, where the highest scoring molecules are likely to be experimentally tested [53]. Recent advancements have led to VS being widely adopted by the industry and academy as a technology of choice in drug discovery [5], including: (i) advancements in 3D structure determination; (ii) improvements in sampling and scoring functions [54]; (iii) implementation of machine learning protocols to computer-assisted drug design; (iv) improvements in computational power, including multiple core design, parallel programming, cloud computing, and GPU processing [55]; and (v) the availability of commercial compounds ready to be checked has strongly increased in recent years. All of these advances have led to the VS approach being proposed as an alternative to experimental HTS, particularly for libraries composed of millions of molecules.

The major goal of the VS technique is the screening speed of large libraries and detection of potential hits during the early stages of drug discovery. The typical approach used in VS is the flexible superposition of a large collection of compounds on a binding pocket of a bioactive molecule (target), evaluating whether or not the ligand contacts will produce any desired effect [56] and providing an accurate picture of the ligand–receptor interaction at the atomic level [57]. The small molecule flexibility can be readily explored in situ or using conformer families. The receptor flexibility is technically more challenging, and it is mandatory to (i) use a small set of protein conformations or (ii) introduce flexibility posteriorly to refine the docking. The receptor ensemble docking uses the merging and shrinking procedure to combine docking results from different 3D receptor structures [58,59]. This method merges the docking results of individual receptor conformations and preserves the best ranked molecule in the ensemble of structures [60,61,62]. The conformations are usually obtained from the structural databases or from molecular dynamic (MD) simulations for sampling. Docking refinements can be accomplished by iteratively changing the interactions between the receptor sidechains and a given molecule ligand, a method usually referred to as soft docking [36,63]. In this procedure, side chain flexibility is simulated by sampling a large number of conformers, allowing for partial clashes with ligand atoms and selecting the most energetically favorable poses. Nevertheless, in both approximations, either the selection of an excessive number or receptor conformers or the use of a large number of side chain conformations has been associated with an increased number of false positive hits and increased computational costs [63,64,65].

To address a VS campaign, a series of steps must be taken, which are schematically described in the Figure 3.

The main steps are:

(i) Structural target selection—Targets of interest are directly taken from the PDB database or modeled to obtain reliable structures for VS. The adequate selection of the structure is crucial for VS results, and failure in this step can condition the docking results. In this respect, it is important to select high-resolution structures, having the conformational state under study, with the adequate apo/holo-protein status. In general, the holo-proteins ensure the correct localization of the binding pocket in an optimal conformation to hold putative binders, and for this reason, these structures are preferred over apo-proteins. Other factors should be considered for target selection and optimization to achieve a successful docking [66]. For example, mutations or incomplete sidechains should be reverted to the wild-type or be rebuilt, especially if located within the ligand site. Missing side chains and loops in the experimental structures should be rebuilt as well if they are close to the binding site, although a more critical rebuilt is needed if these residues present low occupancy, high atomic displacement, or poor electron density maps. Water molecules, cofactors, or metal ions can also be included when the structural resolution allows for it, which are typically those located in the binding pocket or directly interacting with ligands. The protonation state of the protein is critical for the correct determination of the interaction forces. Hydrogen atoms, which are usually unresolved, can be automatically added with reasonable precision, although special care should be taken for residues directly involved in ligand binding [66].

Recently, Stafford et al. developed a method to score a collection of structures based on the docking performance using a set of known active effectors, and the top ranked structures are amenable for use in VS campaigns [67]. The method is limited to targets that have known effector datasets, of which these data are not available for many targets. In the case that the binding pocket is not known, binding site detection is mandatory.

(ii) Binding site prediction—This step is central for structure-based screening, and there are several methods to infer potential binding sites in targets where no ligands have been reported. The methods rely on (a) sequence identity, (b) the reference template used, and (c) geometric and energetic considerations [68]. The sequence-based methods exploit evolutionary information, and the potential binding sites are identified by extracting motif patterns from multiple sequence alignment of already known drug sites. Template-based methods reveal binding sites by comparing them with predefined 3D patterns based on known binding sites. Geometric methods rely on the assumption that a binding site is usually a cleft or a pocket, and they determine complementarity by evaluating the shape, size, and polarity of a binding pocket using putative ligands. ICMPocketFinder (https://www.molsoft.com/icmpocketfinder.html ; accessed on 11 October 2022) is a nice example of a software for determining putative active sites from scratch; It only uses the protein structure for the prediction of cavities and clefts, and no prior knowledge of the substrate is required. The use of this software has allowed for the construction of Pocketome, a database that collects conformational ensembles of druggable binding sites that have been experimentally identified from co-crystal structures in the PDB [69]. Another example is COAH, which generates complementary ligand binding site predictions from given structures of a target protein. It uses two comparative methods, TM-SITE and S-SITE, which recognize ligand-binding templates from the BioLiP database using binding-specific substructure and sequence profile comparisons. The predictions are combined with the program COFACTOR to generate final ligand binding site predictions (https://zhanglab.ccmb.med.umich.edu/COFACTOR/; accessed on 11 October 2022) [70,71]. Additional programs are FPocket [72], MDpocket [73] or SiteMap [74].

As an alternative to the pocket prediction algorithms, docking programs can be used to search for favorable binding sites around the whole protein surface. As an example, CB-Dock is a user-friendly blind docking web server that predicts binding sites of a protein, calculates the centers and sizes of cavities, and performs docking with the popular AutoDock Vina program [75]. Similarly, the EDock program performs blind docking on protein structures whose ligand binding sites are previously predicted by COACH. The initial ligand poses are generated on the predicted binding pockets, and replica-exchange Monte Carlo simulations are performed for conformation sampling using force field and binding site constraints to select the final docking model [76].

Special attention must be paid to proteins containing flexible regions without a pre-formed pocket, commonly referred to as cryptic sites. These sites remain unnoticed in the unbound form, but they are formed after ligand binding, providing a tractable drug target site. It has been suggested that these cryptic sites can provide new sites directed to proteins that would otherwise be considered undruggable [77]. Interestingly, protein–protein interactions include many such cryptic targets that could be potentially used to bind small molecule inhibitors. Furthermore, cryptic sites located away from the orthosteric site of a protein, but with the ability to allosterically modulate the activity of the protein, are potentially useful to improve target specificity [78]. As an example of a cryptic site detection, Cimermancic et al. curated a data set of apo- and holo-protein pairs containing cryptic binding sites to build CryptoSite, a machine learning model to predict such sites in proteins considered undruggable [79].

(iii) Selection of libraries—There are several compound libraries available for use with docking programs in VS campaigns. The bigger the library, the greater the chance of finding more active compounds with favorable pharmacokinetics. The nature of these libraries is diverse, spanning from chemical to natural products, as well as approved drugs, patent-free products, purchasable compounds, etc. Chemical libraries can be generated containing a large number of compounds through several methods, including fragmentation, combination, and deep learning [80]. Nonetheless, natural products, defined as chemicals produced by living organisms, have attracted the attention of the scientific community in the past decade, and its interest continues to incessantly grow. As an example, a collection of open natural products (COCONUT) has been assembled, analyzed, and made available in a user-friendly web interface (https://coconut.naturalproducts.net/; accessed on 14 October 2022). The database is freely available and contains more than 400,000 unique natural products annotated with molecular properties, descriptors, and published biological activities [81]. The ZINC database (https://zinc.docking.org/; accessed on 14 October 2022) contains over 230 million purchasable compounds in ready-to-dock 3D formats as well as 750 million purchasable compounds [82]. ChEMBL is a manually curated database of bioactive molecules that have drug-like properties; it brings together chemical, bioactivity, and genomic data to aid the translation of genomic information into effective new drugs. It contains 2.3 million unique compounds comprising 1.5 million assays, 15,000 targets, and more than 85,000 published papers [83]. Databases such as DrugBank [84] and the Human Metabolome Database [85] are commonly used to repurpose approved drugs to novel targets. Finally, the compound collection Enamine REAL (https://enamine.net/compound-libraries; accessed on 14 October 2022) contains more than 31 billion compounds [86].

(iv) Docking protocol—There are many docking protocols available that deal differently with the flexibility of the molecular structures that intervene in the docking, as stated in Section 3. While rigid docking is used for protein–protein and protein–peptide docking, flexible docking is commonly used for small organic molecules, keeping the receptor rigid or semi-flexible. The selection of the docking applications depends on the system under study, the affordable software resources, and the computational power available. The number of applications for docking is more than 100 [87] and are either free or commercial software. All of the software differs in the conformational search algorithms and in the scoring functions that they are composed of. There are many examples of the use of these applications, and the ability of docking methods to bind ligands into protein has been extensively reviewed [29,30] and adapted for soluble or membrane proteins, including ion channels and receptors [57,88,89]. Nonetheless, there is no one docking application that is superior to others [90].

The most commonly used algorithms for docking are, for example: AutoDock4, which uses an efficient Lamarckian genetic algorithm for global search, a local search for energy optimization, and empirical binding free energy functions [91]; Dock6 uses an anchor-and-grow search algorithm for conformational sampling as well as a footprint similarity scoring function and supports MPI parallelism acceleration [92]; the AutoDock Vina program includes an iterated local search global optimizer, while the binding energy determination combines knowledge-based and empirical scoring functions [93]; PLANTS is another docking program based on an ant colony optimization algorithm to find a minimum energy conformation of the ligand in the protein’s binding site. It uses the empirical scoring functions CHEMPLP and PLP, which are expressly designed for the algorithm [94,95]; RxDock software includes fast intermolecular scoring functions (van der Waals, polar, desolvation), and a stochastic search engine based on a genetic algorithm together with a novel genetic programming-based post-docking filtering to increase the accuracy of the docking [96].

(v) Re-scoring—The poses generated by the docking program are evaluated to find favorable conformations and ranked to select the high scoring hits. Nevertheless, the binding affinity calculation is uncertain because of inherent problems related to the simplified scoring terms. A way to fix this problem is the use of more rigorous energy calculations after the docking process, although it is limited by the excessive computational cost in large chemical libraries. In this way, the results obtained after docking can be rescored and/or filtered by using different scoring functions, such as MLP interactions, which evaluate hydrophobic contacts [97], contacts score, which evaluates interaction of surrounding residues [98], or APBS score, which evaluates ionic interactions [99]. Of note, XScore and DSX are commonly used for rescoring: XScore (https://www.ics.uci.edu/~dock/manuals/xscore1.1_manual/intro.html; accessed on 7 October 2022) computes the binding affinities of the given ligand molecules bound to the target protein by means of an empirical function that comprises van der Waals interactions, hydrogen bonds, and hydrophobic and deformation terms [100,101]. DSX is a knowledge-based scoring function to score protein–ligand complexes of interest and to visualize the per-atom score contributions, which is an intuitive way to learn about differences between putative ligand geometries or learn about the importance of certain binding regions [102]. Either the primary or the recalculated scores can be used to decide which compounds are predicted to bind to the target.

Tran-Nguyen et al. have recently carried out an unbiased evaluation of four scoring functions to rescore docking poses of a high-confidence screening data collection covering several pharmaceutical targets. They found that rescoring based on simplistic knowledge-based scoring functions, e.g., measuring interaction fingerprints, appears to outperform modern machine learning methods, highlighting the importance of the use of rescoring methods to properly detect the most potent binders [103]. Similarly, recent studies have demonstrated that the use of machine learning approaches to rescore docking poses greatly enhances the performance of structural models and that ensembles of rescoring functions increase prediction accuracy [27]. They concluded that the use of empirical data to assess docking predictions is a key factor to improve the prediction of protein–ligand interaction in drug discovery. Finally, Singh et al. have reviewed the structure-based virtual screening web servers, including those having rescoring methods, such as Automatic Molecular Mechanisms Optimization (AMMOS2), CompScore, PlayMolecule, farPPI, and idTarget, which can help not only to identify new hits, but also identify drug repositioning, target-fishing, and polypharmacology prediction (see [104] and references therein). Thus, rescoring techniques can improve the accuracy of the docking results rather than the docking itself being the only filter prior to experimentation [105].

(vi) Post-docking—The structure-based MD approach has been widely used in combination with docking programs to enhance the performance of VS [106]. Although computationally very expensive, MD deals with a spatial and temporal view of the ligand–receptor complex and provides an accurate way to calculate reliable binding affinities [107]. MD is based on the classical equations of motion, estimating the position and the moment of all atoms in the system. The calculated potential energy is used to derive the force acting on all atoms at certain time intervals, resulting in the time evolution of the system as a trajectory [53]. In contrast to the inaccurate binding affinity of docking programs, MD precisely calculates the free energy of the system. There are two main methods: (i) thermodynamic integration, which evaluates the free energy differences for loading/unloading the ligand molecule into the binding pocket versus the bulk; and (ii) alchemical transformations, which pull the ligand from the pocket to the bulk while evaluating the free energy [107,108].

As an alternative, the term post-docking usually refers to low computational cost methods that enhance the hit rates in VS by reducing the number of false positives obtained in docking experiments. As example, the Tanimoto similarity of molecular interaction fingerprints between the predicted and the co-crystal poses accurately discriminates actives from decoys in reference benchmarks [109]. Similarly, machine learning protocols have been used for re-rank docked poses compared with co-crystal ones by the use of a convolutional neural network [67]. Other post-docking strategies have been recently reviewed [40]. The development of consensus models by means of the enrichment factor optimization (EFO) approach is particularly efficient in studies benchmarking the VS docking strategy [110,111].

## 5. Consensus Models of Docking

It has been found that the accuracy of each docking program is system-dependent. This is because the search of best poses depends on the protocols of parameterizations and the training sets used to fine-tune the algorithms. In addition, the performance of VS was shown to greatly vary for the different structural conformations of the proteins, which implies the selection and elimination of structures with the worst performance in the ensemble [112,113]. In the last two decades, several studies have been conducted to evaluate and compare the performance of different docking software on the same systems using known databases of binders and non-binders and using structure ensembles when available [112,113,114,115,116,117,118,119,120,121,122].

Classical consensus approaches focus on the intersection of the best scores produced by individual docking programs [117,118], although more elaborated screening score combinations are commonly used [112,123]. These studies have shown that combining the results of individual docking programs (consensus docking) improves the reliability of the search [124] and helps to obtain a higher success rate in VS [114,115,116,117,118,119,125,126].

### 5.1. Consensus Methods

The screening power measures the ability of programs to identify known binders in a dataset of binders and non-binders (decoys) to determine the agreement between experimental and docking-predicted affinities. There is, however, no single method to quantify this agreement, and several methods to combine the individual results of docking programs have been proposed: rank-by-rank, rank-by-number, rank-by-vote, auto-scales score, Z-score, or exponential consensus ranking score, which were reviewed by Palacio-Rodriguez et al. [112]. In the rank-by-rank method, the rank position of a molecule is obtained as the average rank obtained in each individual docking program [117]. In general, the lower the averaged rank, the better. The rank-by-number calculates the average score over all scoring functions. The rank-by-vote method provides votes to molecules if they are ranked in the top x% of the results of each docking program [117]. Typically, a value of 10% of the benchmark is selected for calculations, where the higher the number of votes, the better. The average auto-scaled scores normalize each docking score between 0 and 1 to avoid differences in scale offset among different docking programs [127]. The rank by Z-score is calculated by subtracting the score of the molecule and the average score of all molecules and dividing by the standard deviation of the scores. The final value is calculated as the average score among all docking programs [128]. The exponential consensus ranking uses an exponential distribution for each rank obtained in the docking runs. An exponential score is calculated for each molecule in all docking programs, and the final score is obtained as the sum of the exponential scores [112]. When available, it is also possible to combine the docking of an ensemble of structures of the protein of interest and the consensus-scoring to determine their impact on the improvements of the VS method [112]. Nevertheless, the prediction accuracy of sampling and scoring power relies on the adequate selection of docking programs and VS workflows [90,113,114,122], and current scoring functions are still not reliable enough [90].

### 5.2. Datasets

The use of molecular datasets where the active compounds are known allows the metric validation to evaluate the performance of the docking methods. In this case, the generation of decoy molecules is mandatory, that is, molecules with physical properties similar to active compounds, but being inactive. The selection of the decoy datasets is not trivial, and several biases have been reported in the literature that may over/underestimate VS performance: analogous, artificial enrichment, and false negative biases [129]. The analogous bias arises in the limited chemical space of the active molecules [130]. The artificial enrichment (or complexity) bias captures the differences in structural complexity between active and decoy molecules [131]. The false negative bias describes the presence of active molecules in the decoy datasets [132].

Property-matched decoys methods match ligands by physical properties (molecular weight, calculated logP, number of rotatable bonds, and hydrogen bond donors and acceptors) but are topologically dissimilar and are presumed not to bind, which represents a challenge for docking programs. In this sense, to avoid mentioned biases, the physicochemical properties between actives and decoys should be well-matched, and the presence of active structures in the decoy sets should be prevented. To further improve the quality of decoy sets, several tools have been developed. The most popular tool to generate decoys is DUD-E [133], a property-matched decoy generator which uses 2D similarity fingerprints to minimize the topological similarity between decoys and actives. Similarly, DEKOIS 2.0 provides a balanced decoy selection to optimize active–decoy physicochemical similarity and to avoid latent actives in the decoy sets [132]. Maximum unbiased validation (MUV) generates data sets with a spatial statistics approach using PubChem HTS bioactivity data [134]. More recently, DeepCoy developed a deep learning method that uses a graph neural network to generate property-matched decoys with user-defined particular requirements [135]. Similarly, TocoDecoy generates unbiased and expandable datasets for training and benchmarking scoring functions based on machine learning. This tool generates property-matched decoy sets in combination with decoy conformation sets having low docking scores to mitigate bias [136]. However, property-matched decoy generation is prone to falsely increase the enrichment and does not represent the chemical space expected in a large library. Stein et al. developed a property-unmatched tool that generates decoy sets that have average physical features of the larger library to be docked, being decoys not too big, not too small, not too hydrophobic, and not too polar [137].

### 5.3. Metric Validation

A parameter commonly used is the enrichment factor (EF), defined as the ratio between the number of actives found in a given percentage of the dataset and the number of compounds at that percentage, normalized by the ratio between the total actives and the total number of compounds in the dataset. EF1% and EF10% are commonly used in VS studies, although they are affected by large variances in datasets with a low number of actives [121]. The confusion matrix allows the visualization of the performance of an algorithm and easily determines whether the system is confusing two classes (binder/non-binder). The binary confusion matrix uses the four kinds of results (TP, true positives, FN, false negatives, FP, false positives, and TN, true negatives) along with the positive and negative classifications. There are many derivations from the confusion matrix, such as sensitivity (recall) or true positive rate [TPR = TP/(TP + FN)], precision [PRE = TP/(TP + FP)], fall-out of false positive rate [FPR = FP/(FP + TN)], specificity or true negative rate [TNR = TN/(TN + FP)], etc.

The calculated metrics can be plotted to discern which docking method is the best. As an example, the enrichment plots (EP) represent the percentage of active compounds recovered in a given percentage of the top-ranked compounds. Similarly, the receiver operating characteristics (ROC), which represents the proportion of TPR against FPR, and the area under the ROC curve (ROC-AUC) are commonly used to evaluate the performance of the model to distinguish binder versus non-binder compounds [121]. Precision–recall (PR) is a curve that combines precision (PRE) and sensitivity (TPR) in a single visualization. The area under the precision–recall curve describes the model performance as the average of the precision scores calculated for each recall threshold. The PR-AUC curve can be used as an alternative metric to evaluate the classifier when the data are imbalanced. In general, PR-AUC provides the ability to differentiate the performance between balanced and imbalanced data and helps to identify the performance around the higher-rank area. Advantages and disadvantages of these evaluation metrics are depicted in Table 1. Other metrics are the robust initial enhancement (RIE) metric, which incorporates an exponential weight as the ranking function [138], and the Boltzmann-Enhanced Discrimination of Receiver Operation Characteristics (BEDROC), a normalized and improved version of RIE [139]. The BEDROC is a metric to quantify early enrichment, thus increasing the contribution from compounds in the first positions. It applies a decreasing exponential function as a weight for the ranking, and adopts values between 0 and 1, which represent the probability that an active, randomly selected, will be better ranked than a compound randomly taken from the database (instead of a uniform distribution as in the ROC). Regarding error estimation, in the context of docking and VS, error has the meaning of “predictive error”, that is, the confidence that the method has to correctly predict, knowing how it has been able to predict a supervised dataset in a retrospective test [121]. In a supervised dataset, the active and inactive compounds are readily known to the researcher, but it is unknown to the docking program itself. The statistics of VS are commonly estimated either numerically (bootstrap) or analytically (formulas) to obtain the error of the metrics. The bootstrapping error is calculated by sampling the original hit list with replacement, generating a set of bootstrapped hit lists. The EF and/or the AUC are then calculated for every hit list, and the error variance is estimated. The analytical error can be computed by calculating the variance for AUC or EF using already described formulas and converting it into 95% confidence interval [121].

There are many examples of the use of consensus docking and the development of metrics to improve docking performance. For instance, Palacio-Rodriguez et al. introduced a novel consensus method based on a sum of exponential distributions as a function of the molecular ranking obtained from each individual program [112]. They evaluated an array of docking programs over four diverse benchmark systems with two target crystallized structures each. They found that the new method outperformed the individual docking results and even the traditional consensus strategies, either using single target or receptor ensemble docking, thus improving the enrichment of actives [112].

Chilingaryan et al. developed a VS workflow integrating several methods and approaches to enhance VS performance. They combined ensemble docking and consensus-scoring approaches in two different scenarios. In the first case, the docking scores were combined between the structures for each docking program, then consensus scoring was applied to rank the compounds. In the second case, the consensus-scoring approach is used for a given structure, then the normalized scores are combined between structures. They concluded that there is a big dependence on the combination of the ensemble and consensus docking used, which increases the VS reliability when correctly used. On the contrary, an inappropriate combination of approaches can lead to a marked decrease in VS performance [113].

Di Stefano et al. identified inhibitors of cyclin-dependent kinase 5 by employing a machine learning-based VS protocol with subsequent molecular docking, molecular dynamics simulations, and binding free energy evaluations. To boost the predictive performance of the VS platform for small-molecule toxicity predictions, they employed a consensus strategy to combine the different predictions of the four top-scored models selected. In this context, the compound was only predicted as active if classified by all four models as active [140].

Gimeno et al. predicted a series of novel inhibitors of the SARS-CoV-2 main protease, a key target for antiviral drugs, through consensus docking and drug reposition. Using two different libraries of approved drugs, they considered bioactive poses that the equivalent high affinity binding modes simultaneously predicted by the three docking programs, taking advantage of the various sampling algorithms without relying on a single scoring function to rank the results [141]. Interestingly, Ochoa et al. developed dockECR, an open computational pipeline for consensus docking and ranking protocols. The protocol uses four open-source molecular docking programs and was calibrated with several protein targets with known actives and decoys. In addition to an exponential consensus method to re-rank molecular candidates, the method employs a scoring strategy based on the average RMSD of best poses from each single program. Using this protocol, the authors evaluated the SARS-CoV-2 main protease and discovered eight inhibitor candidates [142].

Similarly, the DockBox package facilitates the use of multiple docking programs and scoring functions for VS purposes [143]. This package uses score-based consensus docking, which enhanced EF and produced higher hit rates. The approach allows for the use of many scoring functions to assess consensus without a significant computational effort, facilitating the screening of large chemical libraries. Tuccinardi et al. evaluated the reliability of consensus docking by combining ten different docking procedures in terms of consensus cross-docking using an enriched database. The results obtained for three different targets highlighted that consensus docking predicts the ligand binding pose better than the single docking programs and that the VS performs well, substantiating the tenet that this procedure can be used fruitfully for the identification of new hits [144].

## 6. Computational Power

In addition to the progress in structural biology, the refinement of docking algorithms, consensus methods, the set-up of HTVS, or the availability of large chemical libraries, drug discovery has emerged thanks to advancements in computational power, which include (a) grid computing, (b) cloud computing, and (c) hardware acceleration such as graphics processing units [145,146]. Grid computing is composed of several geographically distributed supercomputers, and it can manage massive computational tasks and is widely used in HTVS, thereby reducing time and cost [146]. Cloud computing enables rapid and easy access to shared computing resources, such as networks, servers, storage, applications, etc. [145,147]. The use of accelerators such as graphical processing units (GPUs), used either alone or in conjunction with CPUs, has represented a tremendous computational resource that can currently be used for general purpose computing. GPUs can perform over 500 billion operations per second. As a consequence, molecular modeling applications can be programmed to obtain optimal GPU performance. Several countries have developed E-class computing programs to implement large-scale heterogeneous supercomputing systems. However, although supercomputers correctly manage large files, they fail at handling massive amounts of small molecules files, causing communication pressure on the system.

The best way to solve this issue is currently challenging. Several approaches are commonly used to accelerate VS calculations. As examples, AutoDock4 divides the docking tasks into multiple folders and files and launches individual docking jobs for each ligand. Dock6 supports the message passing interface (MPI) wrapper acceleration by simultaneously launching thousands of docking executions. The MPI is a standardized means of exchanging messages between multiple computers running a parallel program across distributed memory. AutoDock Vina employs multi-thread parallelism, and AutoDock-GPU implements GPU acceleration [148]. These approaches support calculations of up to 10^7^ ligands, but cannot handle ultra-large-scale VS applications. Noteworthily, Zhang et al. have developed aweVS, a package that uses multi-layer databases to integrate all docking tasks and dynamically distribute the large list of docking jobs. The VS process is linearly scaled with the available GPU and CPU to efficiently manage billions of ligands, minimizing the input and output loads [149]. Interestingly, the platform can integrate different docking programs, heterogeneous acceleration software, and different hardware, including GPU processors.

## 7. The Vanilloid Receptor TRPV1: A Case Study

The TRPV1 receptor (or vanilloid receptor) is a polymodal ion channel that responds to both physical and chemical stimuli. TRPV1 is activated by noxious temperatures (≥43 °C), vanilloids (capsaicin), extracellular acidic pH, and membrane depolarization. At the molecular level, TRPV1 is a tetrameric integral membrane protein that presents a modular organization consisting of a transmembrane domain composed of 6 helices (S1 to S6) that form the pore and gating domains, as well as cytosolic domains (N- and C-termini) containing sites for modulating the channel sensitivity [150] (Figure 4). The channel assembly is stable and has been determined by cryo-EM at a 3–4 Å resolution [150,151] in the apo form or in complex with capsaicin, the active principle of chili peppers, which is a desensitizing agonist of TRPV1 [152]. Upon activation, the channel opens and permeabilizes with a preference for Ca^2+^ ions, which activate intracellular signaling pathways. Prolonged exposure to the agonist desensitizes TRPV1 or induces tachyphylaxis to preserve cellular ionic homeostasis. In addition, TRPV1 can be modulated by proinflammatory or pruritogenic agents, and its sensitization produces a notable increase of channel activity due in part to a decrease in the threshold of temperature activation. Other mediators such as NGF increase TRPV1 expression via p28/MAPK or PI3K signaling pathways, thus contributing to TRPV1 sensitization.

These properties are behind the interest of capsaicin as a therapeutic to treat pain and itch [153]. In support of this, capsaicin exhibits clinically relevant analgesic, anti-inflammatory, and anti-pruritic activities targeting TRPV1 [154]. Thus, the central role of the TRPV1 receptor in pain and pruritus has focused the interest of pharmaceutical companies in the discovery of TRPV1 modulators [155,156,157,158]. However, the use of potent TRPV1 antagonists has presented secondary effects (hyperthermia) that have prevented clinical development of the antagonists. In this sense, the discovery of soft inhibitors or activators able to modulate peripheral TRPV1 function with minimal side-effects seems to be a good alternative to alleviate chronic pain and itch [154].

To illustrate a simple VS approach, we used five docking protocols and two re-scoring methods on the TRPV1 mammal receptor as determined by cryo-EM (PDB code: 7LR0) [159] and a dataset of active/inactive compounds. The complex TRPV1–capsaicin was downloaded from the PDB database (RCSB-PDB) and the ligand was stripped-out for the VS study. The library of known inhibitors was obtained from the ChEMBL database [83]. Compounds were classified as active if they had an IC_50_ value of less than 200nM and include “inhibition of capsaicin-induced” in the “Assay Description” heading in ChEMBL. The database filtering produced a total of 212 unique compounds identified as human TRPV1 antagonist, which were considered as the active training set. Decoys were obtained from DUD-E webpage (http://dude.docking.org/; accessed on 2 September 2022), generating 50 decoys per active compound (10,600 inactives) to create a library of 10,812 compounds to screen.

The docking software used was Dock6 [92], AutoDock4 [91], AutoDock Vina [93], PLANTS [94,95], and RxDock [96], keeping the receptor rigid and the ligands free to move. In addition, the AutoDock4 results were re-scored using XScore [100,101] and DSX [102] methods. The docking procedures were performed as homogeneously as possible in order to obtain comparable results. The Dock6 simulation was performed following the geometrical docking procedure implemented in the protocol. The docking search was focused within a 3 Å radius sphere around the reference ligand (capsaicin). The AutoDock4 simulation was accomplished by the Lamarckian genetic algorithm, with the population size being set to 150. Docking scores were calculated by the default scoring function. AutoDock Vina implemented in YASARA used the default optimization parameters for conformational sampling, and the docking scores were calculated with default functions as well. The PLANTS simulation followed the ant colony optimization with a binding site radius of 18 Å from the reference ligand center. Up to 10 poses per ligand were generated and ranked by the ChemPLP scoring function. The RxDock simulation followed the standard docking protocol including three steps of the genetic algorithm search, a Monte Carlo step, and final energy minimization to obtain the best ligand poses. The scoring functions were the default ones. XScore and DSX protocols were used to re-score the sampling performed by AutoDock4 (*dlg* files) with the default empirical scoring functions of these methods was used to determine the binding free energy of the poses (Table 2).

As previously stated, there is sometimes a poor correlation between the results of two different docking programs, as is the case in our TRPV1 docking example. Figure 5A plots the ranking obtained by PLANTS for the selected library compared with the ranking obtained in AutoDock Vina for the same molecules.

The dispersion of the active molecules (red dots) indicates the limitations of the current approaches and the difficulties to encounter the correct poses and/or measure the binding affinity. The active molecules in the upper-left rectangle (Figure 5A) were detected by PLANTS within 10% of the database, but Vina failed to correctly rank them. The contrary was true for the actives in the lower-right rectangle, where PLANTS was unable to correctly find the actives. A classical consensus approach would select molecules in the intersection of both programs (bottom-left square), although many active molecules were lost in the selection. The scenario is more complicated when additional docking programs further restrict the area of intersection to a few active compounds (Figure 5B). As an example, PLANTS and AutoDock Vina detected 37 actives, while PLANTS, AutoDock Vina, and DSX detected 23 actives, or 7 actives if RxDock was additionally included.

To assess the screening accuracy of the docking programs and consensus methods, all results were ranked by score, and the EP and the AUC-ROC were calculated. Figure 6A (left) shows the area under the ROC curve determined for all methods used to dock the library of inhibitors on the selected TRPV1 template. This area indicates how adequate a method is for discriminating active compounds. A value of AUC-ROC equaling 1 means a perfect performance, while a value of 0.5 indicates that the performance of the method does not differ from a random selection of the compounds.

In the present TRPV1 example, the best docking program was AutoDock Vina, which performed similarly or slightly better than the other docking software. On the contrary, Dock6 and AutoDock4 poorly discriminated between binders and non-binders, with AutoDock4 showing the worst docking results. Nonetheless, re-scoring of the poses obtained in AutoDock4 by means of XScore or DSX improved the ability of the model to discriminate between binders and non-binders. The use of consensus strategies (Figure 6A, right) increased the area under the ROC curve, indicating an improvement in the discrimination of the active compounds over decoys. The normalized score ratio (NSR), rank-by-number (RBN), rank-by-rank (RBR) and Z-score methods outperform better than any individual docking software. Figure 6B also illustrates the EP, the percentage of active compounds recovered in a certain percentage of the top-ranked compounds. The shaded area represents the performance of the best and the worst individual docking program compared with the performance of the consensus metrics (colored plots). These consensus metrics produce better results than individual docking programs, thus recovering molecules that were well-ranked by one program but poorly ranked by another, with the global result of performance improvement. Noteworthily, the rank-by-rank (RBR) and exponential consensus ranking (ECR) methods outperform better than other metrics in the 1–10% of the database.

Despite the improvements outlined here, the results are not as good as could be expected. Recently, Llanos et al. performed a structure-based VS on TRPV1 to find potential modulators among approved drugs without thermoregulatory side-effects. Using several TRPV1 structures, they assessed the pose and scoring prediction power to discover three promising candidates for experimental testing [120]. Nonetheless, the worst-performing structure these authors obtained was the capsaicin-derived one, similar to that used in the example. The authors argued that this is something to be expected, as this agonist-derived model was screened with inhibitors, not agonists. In fact, the vanilloid pocket is slightly different in the apo-, agonist-, and antagonist-structures [151,160,161] playing the Y511 orientation, which is an important role in establishing hydrophobic and electrostatic contacts with the effectors [162].

The performance of the VS consensus methods could be easily increased if: (i) a selection of docking protocols and combination of consensus strategies were studied to validate the VS for the actual system; and (ii) several receptor structures were used and compared. In this sense, Chilingaryan et al., searching for inhibitors of dihydroorotate dehydrogenase, concluded that an appropriate combination of ensemble and consensus docking approaches actually increases the reliability of the VS, but an inappropriate combination of these strategies can lead to a dramatic decrease in the performance [113]. Similarly, Manelfi et al., searching for effectors of SARS-CoV-2 protease, determined that the correct combination of the algorithms was key to enhance the VS performance [122].

## 8. Outlook

The understanding of the human genome and the advances experienced by structural biology have identified a large number of proteins likely to be biological targets. The rapid growth of public databases, the fact that these data are capable of being computationally studied, and the increased computing power have boosted the development of computational tools to previously analyze thousands of small molecules to experimental tests. These approaches have the potential to discover drugs with enhanced stability, specificity, and selectivity, attracting the attention of industries and academia.

The computational strategy has a number of advantages, including high speed, low cost, unlimited scalability, and implied automation to limit human intervention, which has revolutionized rational drug design. In contrast, computational approaches require the use of high-resolution atomic structures and optimized methods to evaluate thousands of potential conformations and the theoretical interaction energy between molecules. In this regard, novel and renewed computational methods, such as molecular docking, virtual screening, molecular dynamics, and artificial intelligence, applied together in early stages of the discovery process, may help find clinically useful drugs with improved therapeutic potential and clinical translation.

Molecular docking and VS are the approaches most commonly used to screen large compound libraries. These high-throughput docking methods are very fast to find the most favorable position and orientation of the ligand or to estimate the likelihood of binding, being currently termed as computationally efficient. However, despite recent advances in docking and VS strategies, there are major challenges to be solved: (i) Target availability is a limitation, especially for large membrane protein targets, which traditionally resisted X-ray crystallography. Fortunately, recent advances in the cryo-EM technique have filled that gap by efficiently delivering high-resolution structural models that maintain the functional states of the protein or capturing multiple conformational states in a single experiment [14]. In addition, either homology or ab initio strategies produce high-quality models for performing effective virtual screening, comparable with crystallographic structures, as suggested by several studies [25,26]. (ii) Selection of the active site for docking in the apo-proteins is also challenging as the binding pocket is almost indistinguishable from the rest of the protein surface. The holo-proteins structures facilitate this selection. (iii) The definition of the type of activity for a given ligand is also a threat, and sometimes there are no clues about its possible behavior as activator or inhibitor [163]. (iv) Protein flexibility is challenging because it is not possible to account for backbone and side chain flexibility in an efficient manner. Receptor flexibility is currently limited to a few side chains in the binding pocket. Alternatively, the conformational space of the target can be screened by multiple reference structures [164,165]. In this approach, the results of an ensemble of structures are combined with the merging and shrinking procedure [166,167], which merges the individual docking results of the different structures. Acharya et al. used ensemble docking to obtain target flexibility in docking-based VS on the main protease of SARS-CoV-2 in different protonated states and in monomeric or dimeric form. The docking protocols produced enrichment rates higher than those produced experimentally [168]. (v) The scoring functions for docking are probably the most complex challenge. The docking output presents too many false positives and false negatives, and the hit list critically depends on the quality of the scoring function. Furthermore, different docking software use different sampling strategies and scoring functions, leading to important differences in performance among programs. To date, a large number of studies have been conducted to evaluate and compare docking programs to assess their performance in recovering active molecules [114,144,169,170,171,172,173]. Wang et al. analyzed the results of several docking programs, either public or commercial, more or less popular, newly released or traditional, and determined that although there are clear differences in the scoring functions or conformational search algorithms, there are no significant differences in performance for finding active compounds [114].

Calculating the free energy of protein–ligand binding is not a trivial task. Free energy is a thermodynamic observable that involves (de)solvation effects and entropy changes upon binding, which requires computationally expensive procedures. Nevertheless, the need for a dynamic description of protein–ligand interactions has gradually grown, and several approximations have been developed, mainly based on combined docking and MD strategies, usually referred to as dynamic docking [174]. An MD approach allows for the study of protein–ligand recognition and binding from an energetic and mechanistic point of view, in which the binding and unbinding kinetic constants can be calculated. However, similarly to static docking methods, dynamic docking should generate binding modes, which strongly depends on the sampling strategy used and should evaluate the reliability of the identified poses, processes that are very computationally expensive to be routinely used in drug discovery programs. To solve this drawback, new strategies are being developed, such as the simulation of protein–ligand formation by a guided fast dynamic docking, with the aim of an affordable computational cost [175]. In this sense, it cannot be ruled out that in the future dynamic docking will replace static docking, leading to a paradigm shift in structure-based drug discovery.

In this context, consensus methods have emerged as a tool to improve the reliability of the individual docking programs by overriding the limitations of a single algorithm, enhancing the quality of the predictions from qualitative and quantitative viewpoints. The rationale behind this is that a docking pose with a high consensus level will be more likely to be the biological one. On the contrary, the absence of consensus can be considered suspicious and that the pose should be analyzed and probably discarded. A major challenge in VS simulation is to encounter the combination of search algorithms that correctly predict the binding mode and scoring functions that accurately quantify its binding affinity. Indeed, an added problem is the difference in the effectiveness of the methods in different protein systems. This has been tested in several studies dealing with several protein systems and benchmarks composed of known binders and non-binders [112,113,122], resulting in the conclusion that the correct combination of docking software selection together with the adequate procedure of consensus selection significantly increases the reliability of VS results.

Even though different approaches have been developed to obtain the consensus results from individual docking programs, there is currently no consensus procedure that stands out from the rest of methods. Nevertheless, from the quantitative viewpoint, the analyzed studies strongly support that VS consensus docking can be efficiently used for discovering new hit compounds [112,113,114,115,116,117,118,119,120,122]. The main objection to the consensus approach is probably the computational time, as the compound libraries must be calculated with all docking procedures. To override this inconvenience, alternative strategies are used, such as the hierarchical approach, which employs in a first step the two faster docking procedures. Only the resulting consensus compounds enter in a second step, where the third docking procedure is calculated, and the consensus compounds are re-filtered, and so on [144]. In addition, advances in computational resources have facilitated the computational load. High-performance computing is entering into exascale computing after years of development (https://www.exascaleproject.org/; accessed 28 November 2022.). Quantum computing, the logical evolution of high-performance computing, is a promising tool to be used in drug discovery [176].

The VS methodology applied to the TRPV1 system with pre-validated data produced inconsistencies in the performance of the individual docking processes, although the use of consensus docking protocols slightly improved the global performance. Nevertheless, the use of an ensemble of TRPV1 structures, together with the correct integration of docking methods and metrics, could further improve the outcome performance and reliability of the results, especially when applied to unsupervised libraries.

Overall, consensus docking is a fast, simple, and effective approach that helps the identification of hits in VS campaigns through the identification of the correct poses of ligands bound to the target. Drug discovery can strongly benefit with these promising strategies to bring patients useful drug candidates with improved clinical translation. It is predicted that the consensus docking analysis, together with molecular dynamics simulation and machine learning approaches, will boost the design of new drugs with limited side-effects and enhanced properties, selectivity, and safety.

## Figures and Tables

**Figure 1 molecules-28-00175-f001:**
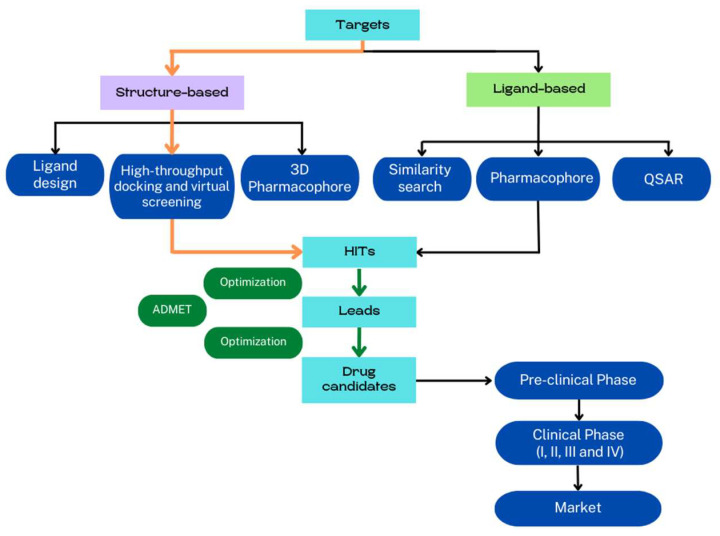
Schematic representation of the drug development cycle. Protocols are mainly classified into structure- and ligand-based methods. Identified hits are optimized to obtain potential candidates for experimental testing. ADMET (absorption, distribution, metabolism, excretion, and toxicity) criteria can be used as additional filters to reduce pre-clinical and clinical attrition rates of potential drugs. The orange arrows indicate the protocols revised.

**Figure 2 molecules-28-00175-f002:**
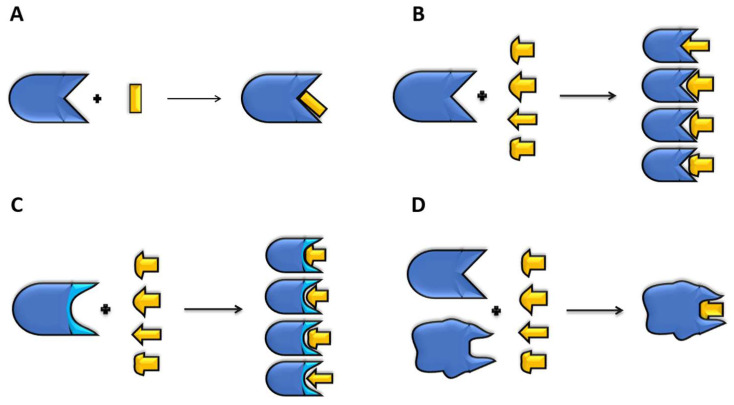
Different levels of docking simplification. (**A**) Rigid docking: neither the receptor (blue) nor ligand (yellow) are allowed to change conformation during the docking process. (**B**) Rigid receptor, flexible ligand: only the small organic molecule (ligand) is allowed to change conformation. Different shapes represent different conformations of the same ligand molecule. The algorithm explores different poses and determines the best score. (**C**) Semi-flexible docking: in addition to the ligand flexibility, the receptor is allowed to change the conformation of a few residues in the binding pocket (cyan) to facilitate ligand interaction. (**D**) Flexible docking: both receptor and ligand are free to change conformation, to improve receptor–ligand matching. Different shapes represent different conformations of the receptor (blue) or ligand (yellow).

**Figure 3 molecules-28-00175-f003:**
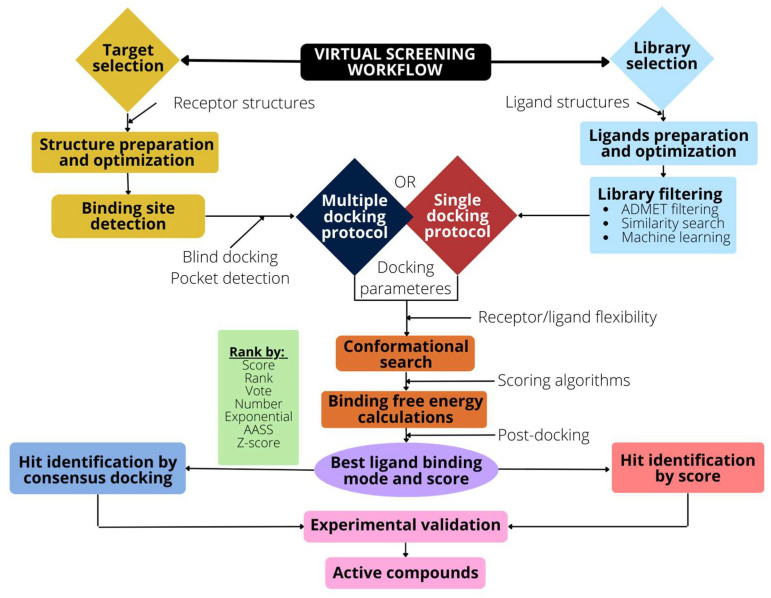
Schematic representation of the VS workflow. The selection and optimization of the target (receptor) is followed by the binding pocket detection and the preparation of the docking input files. Library selection requires optimization and preparation to filter undesired ligands. An ADMET filtering or similarity search can also be used to reduce the ligand space. Docking algorithms are then used to generate poses and scores. The use of a post-docking analysis, such as consensus docking, improves the VS method’s performance.

**Figure 4 molecules-28-00175-f004:**
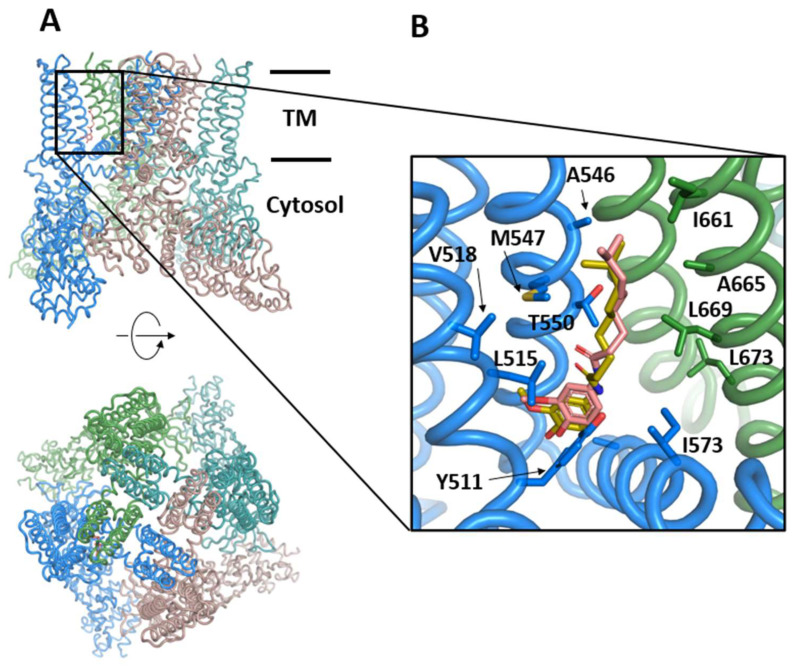
Structure of TRPV1 in the presence of the activator capsaicin, (PDB code: 7LR0). (**A**) Front and top views of TRPV1 in colored illustrations. The horizontal black bars indicate the approximated location of the lipidic membrane and separate the transmembrane (TM) and the cytosolic domains. (**B**) Vanilloid pocket located between two subunits (blue and green) including the capsaicin as determined by cryo-EM (salmon) or re-docked by AutoDock Vina (golden) and the sidechains delineating the pocket.

**Figure 5 molecules-28-00175-f005:**
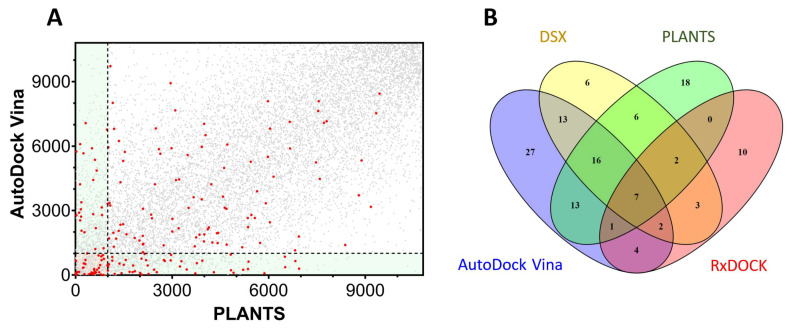
Dispersion of docking results in TRPV1. (**A**) Correlation between the results of AutoDock Vina and PLANTS. The docking results of AutoDock Vina were ranked and plotted against the ranking of the same molecules in PLANTS. The red dots represent the inhibitors obtained from the ChEMBL database while the grey dots indicate the decoys obtained from DUD-E. (**B**) A Venn diagram representing the intersections between four different docking programs (Vina, PLANTS, RxDock and DSX) used on TRPV1. The numbers indicate the amount of shared compounds detected by the different docking methods. The overlapping degree shown was calculated using the top 1000 molecules of the rankings.

**Figure 6 molecules-28-00175-f006:**
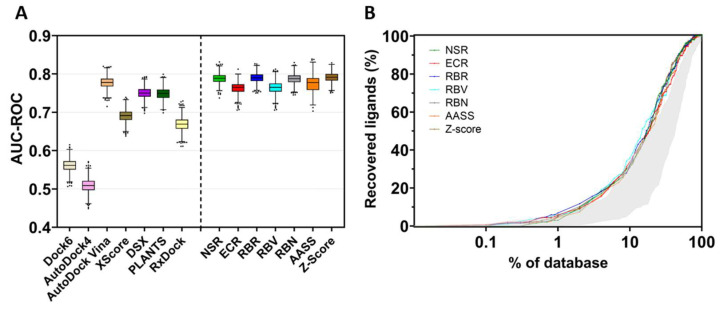
Screening performance of docking programs and consensus strategies. (**A**) area under the curve of the receiver operating characteristics (AUC-ROC). The box plot represents the area obtained for all docking programs and the consensus strategies (NSR: normalized score ratio; ECR: exponential consensus ranking; RBR: rank-by-rank; RBV: rank-by-vote; RBN: rank-by-number: AASS: auto-scaled score; and Z-score). The horizontal line inside the box represents the median value of the distribution, and dots are considered outliers. The mean and the standard deviation were estimated with the bootstrap method (1000 samples). (**B**) Semi-logarithmic representation of EP for docking and consensus strategies. The grey area encompasses the results of the individual docking programs, showing the worst and the best performance. Colored lines represent the performance of the consensus strategies at a given percentage of the database.

**Table 1 molecules-28-00175-t001:** Metric validation. The advantages and disadvantages of receiver operating characteristics (ROC) and precision–recall (PR) evaluation metrics.

Metric	Advantages	Disadvantages
ROC	1. Simple graphical representation and exact measure of the accuracy of a test. 2. Performs equally well on both classes in balanced datasets. 3. The AUC is used as a simple numeric rating of diagnostic test accuracy.	1. Actual decision thresholds are usually not displayed. 2. As the sample size decreases, the plot becomes irregular. 3. Not considered a good indicator for early enrichment of true active samples.
PR	1. Points out the efficiency of the model. 2. Shows how much the data are biased towards one class. 3. Helps understand whether the model is performing well in imbalanced datasets.	1. It does not deal with all the cells of the confusion matrix. True negatives are never considered. 2. Focuses only on positive class. 3. Only suited for binary classification.

**Table 2 molecules-28-00175-t002:** Resources used in the screening workflow for TRPV1.

Stage	Resource	Description
Target structure	RCSB Protein Data Bank (PDB)	PDB is the data center for the global Protein Data Bank (PDB) of 3D structure data for large biological molecules. https://www.rcsb.org; accessed 2 September 2022.
Ligand structures	ChEMBL	ChEMBL is a database of bioactive molecules with drug-like properties. https://www.ebi.ac.uk/chembl/; accessed 2 September 2022.
	Decoys from DUD-E	DUD-E is designed to help benchmark molecular docking programs by providing challenging decoys. http://dude.docking.org; accessed 2 September 2022.
Target preparation	YASARA 22.5.22	YASARA is a molecular modeling and simulation program for structure validation and prediction tools. It is used to rebuild missing side chains and loops. http://www.yasara.org; accessed 1 September 2022.
Ligand preparation	Openbabel 2.4.1	Openbabel. Addition of MMFF94 partial charges, salts removing, protonation at pH 7.4, conversion 2D-3D. https://openbabel.org/docs/dev/Command-line_tools/babel.html; accessed 3 October 2022.
	RDKit 2020.09.1.0	RDKit (Chem package from RDKit). http://www.rdkit.org; accessed 3 October 2022.
	Marvin 6.0	Marvin (molconvert). https://chemaxon.com/marvin; accessed 3 October 2022.
Ligand optimization	RDKit	RDKit (package AllChem). http://www.rdkit.org; accessed 3 October 2022.
	YASARA	YASARA (NOVA force field and energy minimization steps).
ADMET descriptors	Marvin 6	Marvin. ChemAxon’s calculator (cxcalc) is a command line program that performs chemical calculations using calculator plugins. https://chemaxon.com/marvin; accessed 3 October 2022.
	XLOGP3	XLOGP3 is an optimized atom-additive method for the fast calculation of logP. http://www.sioc-ccbg.ac.cn/skins/ccbgwebsite/software/xlogp3/; accessed 6 September 2022.
	RDKit	RDKit is used to obtain molecular descriptors. http://www.rdkit.org; accessed 3 October 2022.
	FILTER-IT	FILTER-IT obtains some molecular descriptors and filters out molecules with unwanted properties. https://github.com/silicos-it/filter-it; accessed 6 September 2022.
	UCSF Chimera 1.15	UCSF Chimera is used for calculations of some molecular descriptors such as SASA and SESA (surf tool). https://www.cgl.ucsf.edu/chimera/; accessed 6 September 2022.
	AMSOL 7.1	AMSOL is used for calculating the free energies of solvation of molecules and ions in solution and partial atomic charges. https://comp.chem.umn.edu/amsol/; accessed 6 September 2022.
Docking	UCSF DOCK6.7	UCSF DOCK6 identifies potential binding geometries and interactions of a molecule to a target using the anchor-and-grow search algorithm. https://dock.compbio.ucsf.edu/DOCK_6/index.htm; accessed 1 September 2022.
	AutoDock4	AutoDock4 performs the docking of the ligands to a set of grids describing the target protein and pre-calculates these grids. https://autodock.scripps.edu; accessed 1 September 2022.
	YASARA	YASARA is used to run macro executing VINA docking algorithms.
	PLANTS	PLANTS is based on ant colony optimization employed to find a minimum energy conformation of the ligand in the protein’s binding site. https://github.com/discoverdata/parallel-PLANTS; accessed 1 September 2022.
	RxDock	RxDock is designed for high-throughput virtual screening campaigns and binding mode prediction studies. https://rxdock.gitlab.io; accessed 1 September 2022.
	XScore	XScore is an empirical scoring function which computes the binding affinities of the given ligand molecules to their target protein. https://www.ics.uci.edu/~dock/manuals/xscore1.1_manual/intro.html; accessed 1 September 2022.
	DSX	DSX is a knowledge-based scoring function that consists of distance-dependent pair potentials, novel torsion angel potentials, and newly defined solvent accessible surface-dependent potentials.
Hits identification (Score-based consensus strategies)	NSR	NSR: Normalized score ratio
	ECR	ECR: Exponential Consensus Ranking
	RBR	RBR: Rank-by-rank
	RBV	RBV: Rank-by-vote
	RBN	RBN: Rank-by-number
	AASS	AASS: Average of auto-scaled score
	Z-Score	Z-Score

## Data Availability

Not applicable.

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
