# Peer review of "Comprehensive Survey of Consensus Docking for High-Throughput Virtual Screening"

_molecules, 2022, doi:10.3390/molecules28010175_

Round 1

Reviewer 1 Report

In this article by Blanes-Mira et al, the authors reviewed the process of consensus docking for high-throughput virtual screening: from the retrieval and processing of input data, to the elements that play important roles in a docking procedure such as search algorithms, scoring functions and evaluation metrics. Overall, this is a well-written and well-organized manuscript, and the authors are to be commended for their effort. However, the authors still need to address the following points in their revision before the manuscript can be accepted for publication.

1. The authors should cite and briefly describe a few published studies that used homology models or AlphaFold-predicted protein structures as templates for prospective structure-based virtual screening. It is interesting and worthwhile to prove that we can achieve good screening results, even prospectively, by using non-experimental target structures, as there are people who still have reservations about the quality of those predicted models and whether they are reliable to do structure-based drug design.

2. Cryptic binding sites can be observed in some protein targets. Is there any docking software/program or published docking study that takes this case into consideration? The authors should provide a few examples if available.

3. The authors should specify that most docking programs use generic scoring functions, which can be applied to any protein target (their applicability domain is large, even not limited). However, past studies have demonstrated that using scoring functions specific to the investigated target leads to better screening results, both retrospectively and prospectively. The authors should discuss this point in the manuscript as well. Is there any docking software/program that integrates different scoring functions, each specific to a certain target or a target family that users can select when the generated docked poses are scored?

4. During the structural target selection process, the authors should mention and discuss other important factors that should be considered, such as the presence of mutations and geometric anomalies in the protein structures.

5. The authors should mention and discuss benchmarking studies that show/prove the benefit (and therefore, highlight the necessity) of re-scoring docked poses generated by a docking program in structure-based virtual screening, for example: https://pubs.acs.org/doi/10.1021/acs.jcim.1c00292.

6. The authors should specify that the decoys provided by DUD-E are property-matched decoys, i.e. decoys whose physico-chemical properties are matched to those of the active molecules. Speaking of this, the authors should also cite more recent tools to generate on-the-fly property-matched decoys, such as DeepCoy (https://academic.oup.com/bioinformatics/article/37/15/2134/6126797) or TocoDecoy (https://pubs.acs.org/doi/10.1021/acs.jmedchem.2c00460). These are machine-learning models that produce high-quality decoys with a lower false negative risk which are deemed more difficult to be distinguished from active molecules. And what about property-unmatched (random) decoys?

7. The authors should also discuss dataset bias (analogue bias, applicability domain bias, potency bias, etc.) that may lead to overestimating virtual screening performance. There are many publications in the literature that addressed this issue.

8. Metric validation: the authors should also mention the area under the precision-recall curve (PR-AUC), which is deemed useful when the classes (active/inactive) are highly imbalanced. The authors should specify the advantages and disadvantages of each evaluation metric in a table. For example, ROC-AUCs are usually not considered a good indicator for early enrichment of true active samples, which is an important factor in virtual screening.

9. The case study of TRPV1: in drug discovery, the ability of a virtual screening method/a scoring function to select different chemical scaffolds among the top-ranked molecules is also very important. Have the authors of this study discussed this point? More discussions are needed.

10. A few minor details need editing, for example: in line 31, it should be "Dhasmana et al refer (or referred) to", not "refers" (no singular verb here).

I am looking forward to receiving the revised version of this manuscript and willing to consider it further for publication.

Author Response

molecules-2097833

Title: Comprehensive evaluation of consensus docking for high-throughput virtual screening

Authors: Clara Blanes-Mira, Pilar Fernández-Aguado, Jorge De Andrés-López, Asia Fernández-Carvajal, Antonio Ferrer-Montiel, Gregorio Fernández-Ballester

Author's Reply to the Review Report (Reviewer 1)

Comments and Suggestions for Authors

In this article by Blanes-Mira et al, the authors reviewed the process of consensus docking for high-throughput virtual screening: from the retrieval and processing of input data, to the elements that play important roles in a docking procedure such as search algorithms, scoring functions and evaluation metrics. Overall, this is a well-written and well-organized manuscript, and the authors are to be commended for their effort. However, the authors still need to address the following points in their revision before the manuscript can be accepted for publication.

Thank you very much for your constructive comments. In the following we will try to answer all your questions and doubts (color blue), including new text added in the revised manuscript (in blue italic).

  1. The authors should cite and briefly describe a few published studies that used homology models or AlphaFold-predicted protein structures as templates for prospective structure-based virtual screening. It is interesting and worthwhile to prove that we can achieve good screening results, even prospectively, by using non-experimental target structures, as there are people who still have reservations about the quality of those predicted models and whether they are reliable to do structure-based drug design.

We have now cited studies combining virtual screening techniques (including consensus strategies), with models derived from homology modelling or ab initio (AlphaFold). Regarding the structures obtained with the artificial intelligence algorithm AlphaFold, there is some controversy. Although AlphaFold has shown impressive results in terms of model accuracy prediction, currently it is not able to distinguish between active and inactive conformations of proteins (see Mullard A. 2021 What does AlphaFold mean for drug discovery? Nat Rev. Drug Discov 20: 725 – 727). There is also a work by Scardino et al. concluding that AlphaFold Models are not reliable enough to be used by VS. They used a benchmark set of 16 different protein targets, four docking programs and two consensus techniques, and found that the AlphaFold models performed consistently worse than their corresponding PDB structures (https://chemrxiv.org/engage/chemrxiv/article-details/6316a14149042ab863cd0481). However, this is a preprint paper, not peer-reviewed, and we prefer not to include it in the manuscript.

The text added at the end of section 2 is as follows. The new references are underlined:

It has been shown that either homology or ab initio models allow effective virtual screening. Several studies have been carried out comparing the performance of homology models and X-ray crystal structures of e.g., G-protein coupled receptors. Carlsson et al. compared the virtual screening results obtained using these scaffolds, and showed that the homology model was as effective as the crystal structure to detect active ligands in terms of hit rate detection, potency and novelty (Carlsson et al 2011. Ligand Discovery from a Dopamine D3 Receptor Homology Model and Crystal Structure. Nat Chem Biol.; 7(11): 769–778). Similarly, Lim et al. found that 10 out of 19 G-protein coupled receptor homology models presented better or comparable performance than the corresponding crystallographic structures, making homology models suitable for virtual screening. They also explored consensus enrichment across multiple homology models, obtaining results comparable to the best performing model, highlighting the usefulness of the consensus scores (Lim et al 2018. A benchmarking study on virtual ligand screening against homology models of human GPCRs. Proteins. 2018;86:978–989). Regarding AlphaFold, several studies have confirmed the suitability of these models to perform reliable VS campaigns. Wong et al. using 12 essential proteins, 218 active and 100 inactive compounds to predict antibacterial inhibitors found that, although models had low performance, the use of rescoring strategies may have acceptable predictive power for certain proteins. They conclude that the limitations in benchmarking are not due to AlphaFold structures itself, but to the methods to accurately model protein-ligand interactions (Wong et al. 2022. Benchmarking AlphaFold-enabled molecular docking predictions for antibiotic discovery. Molecular Systems Biology 18: e11081). Other studies have identified potential inhibitors of WD40 repeat and SOCS box containing 1 protein (WSB1), a clinically relevant drug target, by means of AlphaFold and virtual screening (Weng et al. 2022. Identification of Potential WSB1 Inhibitors by AlphaFold Modeling, Virtual Screening, and Molecular Dynamics Simulation Studies. Evidence-Based Complementary and Alternative Medicine, 2022).”

Also, a small paragraph has been added in the Outlook section:

In addition, either homology or ab initio strategies produce high-quality models in which to perform effective virtual screening, comparable with crystallographic structures, as suggested by several studies. (Carlsson et al 2011; Lim et al 2018)

  1. Cryptic binding sites can be observed in some protein targets. Is there any docking software/program or published docking study that takes this case into consideration? The authors should provide a few examples if available.

There is not, in our knowledge, any software or algorithm combining docking and cryptic sites. These sites are detected experimentally or using computational approaches, such as Markov state models built from hundreds of microseconds of molecular dynamic simulations (see Bowman et al. Equilibrium fluctuations of a single folded protein reveal a multitude of potential cryptic allosteric sites. Proc Natl Acad Sci U S A 2012, 109:11681-11686) to follow the opening and closing of the pocket. However, these results are not free of controversy, and in addition, most of the pockets are not druggable, as discussed by Vajda et al, 2018. Similarly, in the example provided (Cimermancic et al. 2016), the authors initially used 50 features potentially relevant for the cryptic binding site prediction, but only 3 out of 90 proteins proved to be statistically significant.

Nonetheless, the cryptic sites are interesting enough to include a paragraph in section 4 (ii) describing the cryptic sites and the examples. The text added is:

Special attention must be paid to proteins containing flexible regions without pre-formed pocket, commonly referred to as cryptic sites. These sites remain unnoticed in the unbound form, but they are formed after ligand binding, providing a tractable drug target site. It has been suggested that these cryptic sites can provide new sites directed to proteins that would otherwise be considered undruggable (Vajda et al. Cryptic binding sites on proteins: definition, detection, and druggability. Current Opinion in Chemical Biology 2018, 44:1–8). Interestingly, protein-protein interactions include many such cryptic targets that potentially could be used to bind small molecule inhibitors. Furthermore, cryptic sites located away from the orthosteric site of a protein, but with the ability to modulate allosterically the activity of the protein, are potentially useful to improve target specificity (Lu et al. Discovery of hidden allosteric sites as novel targets for allosteric drug design. Drug Discov Today 2018, 23:359-365)”. As an example of cryptic site detection, Cimermancic et al. curated a data set of apo- and holo-protein pairs containing cryptic binding sites to build CryptoSite, a machine learning model to predict such sites in proteins considered undruggable (Cimermancic P; et al. CryptoSite: Expanding the druggable proteome by characterization and prediction of cryptic binding sites. J. Mol. Biol 2016, 428, 709–719).

  1. The authors should specify that most docking programs use generic scoring functions, which can be applied to any protein target (their applicability domain is large, even not limited). However, past studies have demonstrated that using scoring functions specific to the investigated target leads to better screening results, both retrospectively and prospectively. The authors should discuss this point in the manuscript as well. Is there any docking software/program that integrates different scoring functions, each specific to a certain target or a target family that users can select when the generated docked poses are scored?

The generic scoring functions are now commented in section 3.2, indicating that despite its general applicability, targets should be checked previously using appropriate benchmarks. In our knowledge there is no scoring function that can be tuned by the users to adapt to certain types of targets with granted accuracy. Similarly, we don’t know any software allowing automatically the exchange of scoring functions. Anyway, the subject is so extensive that would need a dedicated monograph to encompass all aspects.

The added text is:

Most docking software use generic scoring functions which usually report extensive validation test upon publication, demonstrating their superior performance. It should be pointed out that these functions handle targets unevenly due to certain chemical and structural features, including e.g., the size or exposure of the binding site, the presence of charged groups or the presence of the cofactors/ion metals near the binding site, as well as the protonation state, partial charges and number of rotatable bonds. Thus, it is almost impossible to anticipate the best scoring function for a given target, and the choice commonly relies on the availability of docking software implementing this or another function. The selection of a specific scoring function for a given target involves the design and optimization of a dataset of active/decoys for the specific target. This strategy is subject to the availability of experimental information, but allows a clear and quantitative definition of the limit of validity of the different scoring functions by testing the selected library compounds on the binding site of the selected target (Vieira et al. 2019. Tailoring specialized scoring functions for more efficient virtual screening. Frontiers Drug Chemistry Clinical Res, Vol. 2, 1-4)

Although machine-learning scoring functions have shown superior performance than classical, some cloud of doubts hangs on these scoring functions due to its poor generalization capability and the unfair evaluation of the environment (Shen et al. 2021. Accuracy or novelty: what can we gain from target-specific machine-learning-based scoring functions in virtual screening? Briefings in Bioinformatics, 2021, 1–15). In this sense, the imbalanced datasets, the dataset partitioning or the hidden data biases need to be handled for specific targets. Wallach et al. proposed the asymmetric validation embedding (AVE) strategy to decrease the effects of hidden biases and to avoid similarities between validation and training datasets, which represents an important tool to evaluate the general applicability of the scoring function based on machine-learning (Wallach et al. 2018. Most ligand-based classification benchmarks reward memorization rather than generalization. J Chem Inf Model 2018;58:916–32).”

  1. During the structural target selection process, the authors should mention and discuss other important factors that should be considered, such as the presence of mutations and geometric anomalies in the protein structures.

Thanks a lot for this comment. Details of this section were omitted in the previous version for simplicity. Some other factors involved in target selection and optimization are now briefly commented in section 4 (i). The paper Bender et al. A practical guide to large-scale docking. Nature Protocols. Vol 16. 2021 is now referenced. The text added is:

“Other factors should be considered for target selection and optimization to achieve a successful docking (Bender et al. A practical guide to large-scale docking. Nature Protocols. Vol 16. 2021). For example, mutations or incomplete sidechains should be reverted to wild type or rebuilt, especially if located within the ligand site. Missing side chains and loops in the experimental structures should be rebuilt as well if they are close to the binding site, although a more critical rebuilt is needed if these residues present low occupancy, high atomic displacement or poor electron density maps. Water molecules, cofactors or metal ions can also be included when the structural resolution allows it, typically those located in the binding pocket or interacting directly with ligands. The protonation state of the protein is critical for the correct determination of interaction forces. Hydrogen atoms, that are usually unresolved, can be added automatically with reasonable precision, although special care should be taken for residues directly involved in ligand binding (Bender et al. 2021).

  1. The authors should mention and discuss benchmarking studies that show/prove the benefit (and therefore, highlight the necessity) of re-scoring docked poses generated by a docking program in structure-based virtual screening, for example: https://pubs.acs.org/doi/10.1021/acs.jcim.1c00292.

Thanks a lot for the suggested reference. It has certainly been shown that rescoring poses play an important role in virtual screening campaigns. We have now referenced the paper suggested, as well as other recent works, including a review describing web servers able to rescore docking poses.

The text added in section 4 (v) is:

“Tran-Nguyen et al. have recently carried out an unbiased evaluation of four scoring functions to rescore docking poses of a high-confidence screening data collection covering several pharmaceutical targets. They found that rescoring based on simplistic knowledge-based scoring functions, e.g., measuring interaction fingerprints, appears to outperform modern machine learning methods, highlighting the importance of the use of rescoring methods to properly detect the most potent binders (Tran-Nguyen et al. 2021. True Accuracy of Fast Scoring Functions to Predict High-Throughput Screening Data from Docking Poses: The Simpler the Better. J. Chem. Inf. Model. 2021, 61, 2788−2797). Similarly, recent studies demonstrated that the use of machine-learning approaches to rescore docking poses greatly enhances the performance of structural models, and that ensembles of rescoring functions increases prediction accuracy (Wong et al. 2022. Benchmarking AlphaFold-enabled molecular docking predictions for antibiotic discovery. Molecular Systems Biology 18: e11081). They concluded that the use of empirical data to assess docking predictions is a key factor to improve the prediction of protein-ligand interaction in drug-discovery. Finally, Singh et al. have reviewed the structure-based virtual screening web servers, including those having rescoring methods, such as Automatic Molecular Mechanisms Optimization (AMMOS2), CompScore, PlayMolecule, farPPI, and idTarget that can help not only to identify new hits, but also to drug repositioning, target-fishing, and polypharmacology prediction (see Singh et al. 2021. Virtual screening web servers: designing chemical probes and drug candidates in the cyberspace. Briefings in Bioinformatics, 22(2), 2021, 1790–1818 and references therein). Thus, rescoring techniques can improve the accuracy of docking results, rather than being the docking itself the only filter prior to experimentation (Glaser et al. 2021. High-throughput virtual laboratory for drug discovery using massive datasets. The International Journal of High Performance Computing Applications 1–17).”

  1. The authors should specify that the decoys provided by DUD-E are property-matched decoys, i.e. decoys whose physico-chemical properties are matched to those of the active molecules. Speaking of this, the authors should also cite more recent tools to generate on-the-fly property-matched decoys, such as DeepCoy (https://academic.oup.com/bioinformatics/article/37/15/2134/6126797) or TocoDecoy (https://pubs.acs.org/doi/10.1021/acs.jmedchem.2c00460). These are machine-learning models that produce high-quality decoys with a lower false negative risk which are deemed more difficult to be distinguished from active molecules. And what about property-unmatched (random) decoys?

Section 5.2 has been completely rephrased to use specifically “property-matched decoy” to describe DUD-E tool. In addition, we have referenced other tools such as, DEKOIS 2.0, MUV, and the more recent tools DeepCoy, TocoDecoy, and also a tool that generates property-unmatched decoys, as suggested. This section has been completed with bias description (see next paragraph #7 to authors)

Added paragraph at the end of section 5.2:

“Property-matched decoys methods match ligands by physical properties (molecular weight, calculated logP, number of rotatable bonds, and hydrogen bond donors and acceptors) but are topologically dissimilar, and are presumed not to bind, which represents a challenge for docking programs. In this sense, to avoid mentioned biases, the physicochemical properties between active and decoys should be well matched, and the presence of active structures in the decoy sets should be prevented. To further improve the quality of decoy sets, several tools have been developed. The most popular tool to generate decoys is DUD-E [115], a property-matched decoy generator which uses 2-D similarity fingerprints to minimize the topological similarity between decoys and actives. Similarly, DEKOIS 2.0 provides a balanced decoy selection to optimize active-decoy physicochemical similarity and to avoid latent actives in the decoy sets (Bauer et al. Evaluation and Optimization of Virtual Screening Workflows with DEKOIS 2.0 − A Public Library of Challenging Docking Benchmark Sets. J. Chem. Inf. Model. 2013, 53, 1447−1462. 2013). Maximum Unbiased Validation (MUV) generates data sets with a spatial statistics approach using PubChem HTS bioactivity data (Rohrer, S. G.; Baumann, K. Maximum unbiased validation (MUV) data sets for virtual screening based on PubChem bioactivity data. J. Chem. Inf. Model. 2009, 49, 169−184). More recently, DeepCoy developed a deep-learning method that uses a graph neural network to generate property-matched decoys with user-defined particular requirements (Imrie et al. Generating property-matched decoy molecules using deep learning. Bioinformatics, 37(15), 2021, 2134–2141. 2021). Similarly, TocoDecoy generates unbiased and expandable datasets for training and benchmarking scoring functions based on machine-learning. This tool generates property-matched decoy sets in combination with decoy conformation sets with low docking scores, to mitigate bias (Zhang et al. TocoDecoy: A New Approach to Design Unbiased Datasets for Training and Benchmarking Machine-Learning Scoring Functions. J. Med. Chem. 2022, 65, 7918−7932. 2022). However, property-matched decoy generation is prone to increase the enrichment falsely and does not represent the chemical space expected in a large library. Stein et al. developed a property-unmatched tool that generates decoy sets that have average physical features of the larger library to be docked, being decoys not too big, not too small, not too hydrophobic, and not too polar (Stein et al. Property-Unmatched Decoys in Docking Benchmarks. Chem Inf Model. 2021 February 22; 61(2): 699–714. 2022).”

  1. The authors should also discuss dataset bias (analogue bias, applicability domain bias, potency bias, etc.) that may lead to overestimating virtual screening performance. There are many publications in the literature that addressed this issue.

The dataset biases description has been added at the beginning of section 5.2, and referenced. This section has been completed with tools description to generate decoy sets (see previous paragraph #6 to authors).

Added text in the manuscript is:

The use of molecular datasets where the active compounds are known, allows the metric validation to evaluate the performance of the docking methods. In this case the generation of decoy molecules is mandatory, that is, molecules with physical properties similar to active compounds, but being inactive. The selection of the decoy datasets is not trivial, and several biases have been reported in the literature that may over/underestimate VS performance: analogous, artificial enrichment and false negative biases (Reau et al, 2018. Decoys Selection in Benchmarking Datasets: Overview and Perspectives. Frontiers in Pharmacology. 9. 11; Sieg,J. et al. 2019. In need of bias control: evaluating chemical data for machine learning in structure-based virtual screening. J. Chem. Inf. Model., 59, 947–961). The analogous bias arises in the limited chemical space of the active molecules (Good, A. C., and Oprea, T. I. 2008. Optimization of CAMD techniques 3. Virtual screening enrichment studies: a help or hindrance in tool selection? J. Comput. Aided Mol. Des. 22, 169–178). The artificial enrichment (or complexity) bias captures the differences in structural complexity between active and decoy molecules (Stumpfe, D., and Bajorath, J. (2011). “Applied virtual screening: strategies, recommendations, and caveats,” in Virtual Screening: Principles, Challenges, and Practical Guidelines, ed C. Sotriffer (Weinheim:Wiley-VCH Verlag GmbH and Co. KGaA), 291–318). The false negative bias describes the presence of active molecules in the decoy datasets (Bauer, M. R., Ibrahim, T. M., Vogel, S. M., and Boeckler, F. M. (2013). Evaluation and optimization of virtual screening workflows with DEKOIS 2.0 – a public library of challenging docking benchmark sets. J. Chem. Inf. Model. 53, 1447–1462).

  1. Metric validation: the authors should also mention the area under the precision-recall curve (PR-AUC), which is deemed useful when the classes (active/inactive) are highly imbalanced. The authors should specify the advantages and disadvantages of each evaluation metric in a table. For example, ROC-AUCs are usually not considered a good indicator for early enrichment of true active samples, which is an important factor in virtual screening.

The receiver operating characteristic (ROC) curve and the precision-recall (PR) curve are visual tools for comparing binary classifiers, and the area under the ROC curve or under the PR curve summarizes the ROC and PR curves in numbers. Although we agree with the referee that it is commonly argued that the PR curve is preferred over the ROC curve for imbalanced data, it should be pointed out that it depends on the specific application context. ROC can be misleading or uninformative, and can contain less truth than assumed, but PR curve can equally well disguise important aspects of prediction accuracy and be misleading when there is class imbalance, and the response variable is difficult to predict. In addition, if the ROC curve of one classifier is always above the ROC curve of another classifier, the same also holds true for the PR curve, and vice versa (https://dl.acm.org/doi/10.1145/1143844.1143874).

A brief description of the Precision-Recall metric has been added to the section 5.3, as well as a new Table 1 containing advantages and disadvantages of ROC and PR curves. Parameter precision (PRE) has been also defined in the previous paragraph.

Text added is:

… non-binders compounds [107]. Precision-Recall (PR-AUC) is a curve that combines precision (PRE) and sensitivity (TPR) in a single visualization. The area under the Precision-Recall Curve describes model performance as the average of precision scores calculated for each recall threshold. PR-AUC curve can be used as an alternative metric to evaluate the classifier when the data is imbalanced. In general, PR-AUC provides the ability to differentiate the performance between balanced and imbalanced data, and helps to identify the performance around higher-rank area. Advantages and disadvantages of these evaluation metrics are depicted in Table 1.”.

And the Table 1:

“Table 1. Metric validation. Advantages and disadvantages of Receiver Operating Characteristics (ROC) and Precision-Recall (PR) evaluation metrics.”

Metric

Advantages

Disadvantages

ROC

1. Simple graphical representation and exact measure of the accuracy of a test.

2. Performs equally well on both classes in balanced datasets.

3. The AUC is used as a simple numeric rating of diagnostic test accuracy.

1. Actual decision thresholds are usually not displayed.

2. As the sample size decreases, the plot becomes irregular.

3. Not considered a good indicator for early enrichment of true active samples.

PR

1. Points out the efficiency of the model.

2. Shows how much the data is biased towards one class.

3. Helps understand whether the model is performing well in imbalanced datasets.

1. It does not deal with all the cells of the confusion matrix. True negatives are never considered.

2. Focuses only on positive class.

3. Suited only for binary classification.

  1. The case study of TRPV1: in drug discovery, the ability of a virtual screening method/a scoring function to select different chemical scaffolds among the top-ranked molecules is also very important. Have the authors of this study discussed this point? More discussions are needed.

The case study was just a quick, easy and simple example of inhibitor searching to illustrate that the consensus scoring analysis outperforms single scoring to detect hits. Nevertheless, in the manuscript we highlight the importance of TRPV1 modulation to treat pain and itch, and we present a VS approach using a single TRPV1 structure. We already comment in the manuscript that the VS results are not good enough in terms of enrichment, probably due to the fact that the selected TRPV1 structure conformation corresponds to an agonist (capsaicin) and the screened library is a selection of inhibitors. In addition, we outline the steps that should be done to increase performance, such as the selection of docking protocols, the use of different libraries, or the use of several TRPV1 structures (section 7). Indeed, we are currently conducting docking experiments on different TRPV1 scaffolds, using additional scoring protocols, and different rescoring methods, together with the consensus strategies to select appropriate chemical scaffolds likely to modulate TRPV1. We agree with the referee this is quite interesting. Nevertheless, we believe this work deserves a dedicated study carefully describing and discussing the results obtained.

  1. A few minor details need editing, for example: in line 31, it should be "Dhasmana et al refer (or referred) to", not "refers" (no singular verb here).

The error has been corrected, as well as other minor errors. In addition, authors referenced in the text as “et al” have now the final period: “et al.”.

I am looking forward to receiving the revised version of this manuscript and willing to consider it further for publication.

Reviewer 2 Report

Dec. 09, 2022

Review on “Comprehensive evaluation of consensus docking for high-throughput virtual screening” by Clara Blanes-Mira, Pilar Fernández-Aguado, Jorge de Andrés-López, Asia Fernández-Carvajal, Antonio Ferrer-Montiel and Gregorio Fernández-Ballester

The authors present an excellent summary of current structure-based virtual screening approaches. Several methods at each stage of screening were comprehensively reviewed, providing underlying ideas, algorithms, tools, potential problems. This review certainly helps readers to grasp the current status of the structure-based virtual screening, and to design their own protocols for screening. The manuscript is well written and clear enough for the publication. It can be published as is, but I suggest authors to consider the following points to improve their manuscript.

·        I was a little confused by the term “evaluation” in the title. I thought this paper was more “survey” than “evaluation”.

·        Authors could benefit from adding a paragraph to the discussion of “dynamic docking” using molecular dynamics (MD) simulation. The author first introduced MD in the section for search algorithms in p5. Given the rapid growth of “dynamic docking”, an additional statement may be of value to the reader. Some useful references are given below:

(1)   De Vivo, Marco, Matteo Masetti, Giovanni Bottegoni, and Andrea Cavalli. “Role of Molecular Dynamics and Related Methods in Drug Discovery.” Journal of Medicinal Chemistry 59, no. 9 (2016): 4035–61.

(2)   Gioia, Dario, Martina Bertazzo, Maurizio Recanatini, Matteo Masetti, and Andrea Cavalli. “Dynamic Docking: A Paradigm Shift in Computational Drug Discovery.” Molecules 22, no. 11 (2017): 1–21. 

·        It would be great if authors could provide a table listing the tools, packages, databases presented at each stage of the screening workflow.

·        In Figure 1, "Structure-based" and "Ligand-based" represent classifications and are distinct from "target", "HITs", etc., and are therefore recommended to be shown with different background colors.

·        In Figure 2, it is easier to understand if authors specify where the post-docking corresponds. Is the “Binding free energy calculations” corresponding to the post-docking stated in the main text ((vi) in p10)? On the top-left, “Receptor/s structure/s” may be changed to “Receptor structures” for consistency. 

Author Response

Author's Reply to the Review Report (Reviewer 2)

Open Review

Comments and Suggestions for Authors

Dec. 09, 2022

Review on “Comprehensive evaluation of consensus docking for high-throughput virtual screening” by Clara Blanes-Mira, Pilar Fernández-Aguado, Jorge de Andrés-López, Asia Fernández-Carvajal, Antonio Ferrer-Montiel and Gregorio Fernández-Ballester

The authors present an excellent summary of current structure-based virtual screening approaches. Several methods at each stage of screening were comprehensively reviewed, providing underlying ideas, algorithms, tools, potential problems. This review certainly helps readers to grasp the current status of the structure-based virtual screening, and to design their own protocols for screening. The manuscript is well written and clear enough for the publication. It can be published as is, but I suggest authors to consider the following points to improve their manuscript.

Thank you very much for your positive comments. Below we answer your questions and suggestions (color blue). We also include the text added to the manuscript (blue italic).

  • I was a little confused by the term “evaluation” in the title. I thought this paper was more “survey” than “evaluation”.

The word “evaluation” in the title has been changed by “survey”.

  • Authors could benefit from adding a paragraph to the discussion of “dynamic docking” using molecular dynamics (MD) simulation. The author first introduced MD in the section for search algorithms in p5. Given the rapid growth of “dynamic docking”, an additional statement may be of value to the reader. Some useful references are given below:

(1)   De Vivo, Marco, Matteo Masetti, Giovanni Bottegoni, and Andrea Cavalli. “Role of Molecular Dynamics and Related Methods in Drug Discovery.” Journal of Medicinal Chemistry 59, no. 9 (2016): 4035–61.

(2)   Gioia, Dario, Martina Bertazzo, Maurizio Recanatini, Matteo Masetti, and Andrea Cavalli. “Dynamic Docking: A Paradigm Shift in Computational Drug Discovery.” Molecules 22, no. 11 (2017): 1–21.

Thanks a lot for the suggested references. Molecular dynamics was introduced in section 3.1, and revisited in section 4.(vi) Post-docking to highlight the more precise calculation of binding energy by means of MD.

As suggested by the referee, we added the following paragraph in the Outlook section. The new references are underlined:

Calculating free energy of protein-ligand binding is not a trivial task. Free energy is a thermodynamic observable that involves (de)solvation effects and entropy changes upon binding, which requires computationally expensive procedures. Nevertheless, the need for a dynamic description of protein-ligand interactions has grown gradually, and several approximations have been developed, mainly based on combined docking and MD strategies, usually referred to as dynamic docking (Gioia et al. 2017. "Dynamic Docking: A Paradigm Shift in Computational Drug Discovery" Molecules 22, no. 11: 2029.; De Vivo et al. 2016. Role of Molecular Dynamics and Related Methods in Drug Discovery. J Med Chem. 2016 May 12;59(9):4035-61). MD allows the study of protein-ligand recognition and binding from an energetic and mechanistic point of view, in which the binding and unbinding kinetic constants can be calculated. However, in analogy to static docking methods, dynamic docking should generate binding modes, which strongly depends on the sampling strategy used, and should evaluate the reliability of the identified poses, processes that are very computationally expensive to be used routinely in drug discovery programs. To solve this drawback, new strategies are being developed, such as the simulation of protein-ligand formation by a guided fast dynamic docking, at affordable computational cost (Spitaleri et al. 2018. Fast Dynamic Docking Guided by Adaptive Electrostatic Bias: The MD-Binding Approach. Journal of Chemical Theory and Computation 2018 14 (3), 1727-1736). In this sense, it cannot be ruled out that in the future dynamic docking will replace static docking, leading to a paradigm shift in structure-based drug discovery.

  • It would be great if authors could provide a table listing the tools, packages, databases presented at each stage of the screening workflow.

We have included a table (Table 2) containing the tools used at different stages of VS that we used in the case study of TRPV1. It is added in section 7 and referenced in the text. We hope this is the list suggested by the referee.

The table added is:

“Table 2. Resources used in the screening workflow for TRPV1.”

STAGE

RESOURCE

DESCRIPTION

Target structure

RCSB Protein Data Bank (PDB)

PDB is the data center for the global Protein Data Bank (PDB) of 3D structure data for large biological molecules. https://www.rcsb.org

Ligand structures

ChEMBL

Decoys from DUD-E

ChEMBL is a database of bioactive molecules with drug-like properties. https://www.ebi.ac.uk/chembl/

DUD-E is designed to help benchmark molecular docking programs by providing challenging decoys. http://dude.docking.org

Target preparation

YASARA 22.5.22

YASARA is a molecular modeling and simulation program for structure validation and prediction tools. It is used to rebuild missing side chains and loops. http://www.yasara.org

Ligand preparation

Openbabel 2.4.1

RDKit 2020.09.1.0

Marvin 6.0

Openbabel. Addition of MMFF94 partial charges, salts removing, protonation at pH 7.4, conversion 2D-3D. https://openbabel.org/docs/dev/Command-line_tools/babel.html

RDKit (Chem package from RDKit). http://www.rdkit.org

Marvin (molconvert).; https://chemaxon.com/marvin

Ligand optimization

RDKit

YASARA

RDKit (package AllChem). http://www.rdkit.org

YASARA (NOVA Force Field and energy minimization steps).

ADMET descriptors

Marvin 6

XLOGP3

RDKit

FILTER-IT

UCSF Chimera 1.15

AMSOL 7.1

Marvin. ChemAxon's Calculator (cxcalc) is a command line program that performs chemical calculations using calculator plugins. https://chemaxon.com/marvin

XLOGP3 is an optimized atom-additive method for the fast calculation of logP. http://www.sioc-ccbg.ac.cn/skins/ccbgwebsite/software/xlogp3/

RDKit is used to obtain molecular descriptors. http://www.rdkit.org

FILTER-IT obtains some molecular descriptors and filters out molecules with unwanted properties. https://github.com/silicos-it/filter-it

UCSF Chimera is used for calculations of some molecular descriptors as SASA and SESA (surf tool). https://www.cgl.ucsf.edu/chimera/

AMSOL is used for calculating free energies of solvation of molecules and ions in solution and partial atomic charges. https://comp.chem.umn.edu/amsol/

Docking

UCSF DOCK6.7

AutoDock4

YASARA

PLANTS

RxDock

XScore

DSX

UCSF DOCK6 identifies potential binding geometries and interactions of a molecule to a target using the anchor-and-grow search algorithm. https://dock.compbio.ucsf.edu/DOCK_6/index.htm

AutoDock4 performs the docking of the ligands to a set of grids describing the target protein, and pre-calculates these grids. https://autodock.scripps.edu

YASARA is used to run macro executing VINA docking algorithms.

PLANTS is based on ant colony optimization employed to find a minimum energy conformation of the ligand in the protein’s binding site. https://github.com/discoverdata/parallel-PLANTS

RxDock is designed for high-throughput virtual screening campaigns and binding mode prediction studies. https://rxdock.gitlab.io

XScore is an empirical scoring function which computes the binding affinities of the given ligand molecules to their target protein. https://www.ics.uci.edu/~dock/manuals/xscore1.1_manual/intro.html

DSX is a knowledge-based scoring function that consists of distance-dependent pair potentials, novel torsion angel potentials, and newly defined solvent accessible surface-dependent potentials.

Hits identification (Score-based consensus strategies)

NSR

ECR

RBR

RBV

RBN

AASS

Z-Score

NSR: Normalized score ratio

ECR: Exponential Consensus Ranking

RBR: Rank-by-rank

RBV: Rank-by-vote

RBN: Rank-by-number

AASS: Average of auto-scaled score

Z-Score

  • In Figure 1, "Structure-based" and "Ligand-based" represent classifications and are distinct from "target", "HITs", etc., and are therefore recommended to be shown with different background colors.

Figure 1 has been changed according to the suggestion. Structure-based and Ligand-based have now different background color accordingly.

  • In Figure 2, it is easier to understand if authors specify where the post-docking corresponds. Is the “Binding free energy calculations” corresponding to the post-docking stated in the main text ((vi) in p10)? On the top-left, “Receptor/s structure/s” may be changed to “Receptor structures” for consistency.

We understand that the referee refers to Figure 3 and not to Figure 2. Once the docking procedures (conformational search and binding free energy calculations) are finished, the post-docking methods start, including consensus strategies, MD, etc. We have included an arrow indicating the starting point of the post-docking analysis. We have also added the label “Receptor structures” on the top.

Round 2

Reviewer 1 Report

This revised version of the manuscript by Blanes-Mira et al addressed my previous comments and is therefore much improved. I am happy to recommend it for publication in Molecules, and congratulate the authors on the completion of this good work. Well done!